# Nature-inspired hierarchical building materials with low CO$_2$ emission and superior performance

Jinyang Jiang[1], Han Wang[1], Junlin Lin[1], Fengjuan Wang[1], Zhiyong Liu[1], Liguo Wang[1], Zongjin Li[2], Yali Li[3], Yunjian Li [2] ✉ & Zeyu Lu [1] ✉

Conventional cement-based materials are faced with significant challenges, including large carbon emissions, high density, and quasi-brittleness. Here, inspired by hierarchical porous structures existing in nature, we develop a low carbon, lightweight, strong and tough cement-based material (LLST), which is obtained by a rapid gelation of hydrogel as skeleton and subsequent deposition of cement hydrates as a skin. As a result, the LLST exhibits hierarchical structure consisting of sponge-like micropores (1 ~ 50 μm) and nanopores (5 ~ 100 nm), without detrimental macropores that compromise light weight, strength, and toughness. Compared with the normal cement paste, LLST displays a 54% reduction in density, 145% and 1365% improvement in specific compressive strength and fracture energy, with only 51% carbon emission. These properties are further investigated with machine learning force field molecular dynamics along with well-tempered metadynamics simulations, indicating that strong chemical bonding is generated at the atomic level between functional groups in the hydrogel and Ca ion released from cement hydration. These findings not only demonstrate a strategy for developing lightweight building materials with low-carbon emission and remarkable mechanical properties, but also provide valuable insights for realizing the coexistence of light weight, strength and toughness by tailoring the hierarchical pore structure.

The reduction of carbon dioxide (CO$_2$) emissions has gained widespread recognition as a crucial approach to addressing the climate crisis and realizing environmental sustainability. Particularly, cement industry causes 7% of global CO$_2$ emissions, and is ranked the third largest source after power and steel industries[1]. As a result, it becomes imperative to develop low-carbon cement-based materials to meet the carbon neutrality target set in the Paris Agreement[2]. Recently, the development of lightweight cement-based materials with enhanced strength and toughness has been recognized as a promising strategy for reducing CO$_2$ emissions in the cement industry by decreasing

cement consumption, extending service life, and lowering energy demand in production and transportation, etc[3–5].

However, current lightweight design for cement-based materials is usually at the expense of strength and toughness[6–8]. For instance, foam technology is commonly used for the preparation of lightweight cement-based materials, in which large amounts of macropores are introduced into cement matrix thus lowering the density from 2.2 ~ 2.4 g/cm$^3$ to 0.01 ~ 1.2 g/cm$^3$. However, the stress concentration caused by large pores in cement matrix leads to a significantly sacrificed mechanical performance (compressive strength: 0.2 ~ 10 MPa,

[1]State Key Laboratory of Engineering Materials for Major Infrastructure, School of Materials Science and Engineering, Southeast University, Nanjing, China. [2]Faculty of Innovation Engineering, Macau University of Science and Technology, Macau, SAR, China. [3]Centre for Smart Infrastructure and Digital Construction, School of Engineering, Swinburne University of Technology, Hawthorn, Victoria, Australia. ✉e-mail: liyunjian@must.edu.mo; 101012819@seu.edu.cn

flexural strength: 0.05 ~ 4 MPa)[9]. Ice-template technique is an alternative approach for preparing lightweight (0.01 ~ 0.8 g/cm[3]) cement-based materials with smaller pore sizes ranging from 10 ~ 500 μm[10,11]. The regularly arranged smaller pores contribute to much improved compressive strength (20 ~ 50 MPa) and toughness (10 ~ 14 MJ·m[−3]) along the direction of ice crystal growth, but unfortunately a 300% ~ 500% reduction in mechanical strength in the transverse direction[12], owing to the anisotropic nature of materials fabricated by ice-template method[13]. Therefore, there is a need for a novel strategy to consolidate the lightweight, strong and toughness properties of cement-based materials.

Nature provides an ever-expanding source of inspiration and prospects for the design of lightweight materials with high strength and toughness through tailoring hierarchically porous structure[14–16]. Cancellous bone, honeycomb and coral, as typical hierarchically porous materials with the feature of sponge-like structure[17,18], exhibit exceptional lightweight and mechanical properties due to their fine-grained pore structure, which helps to disperse concentrated stress and absorb energy under external loading. Recently, a series of lightweight materials (0.8 ~ 1.5 g/cm[3]) with remarkable mechanical strength (compressive strength: 20 ~ 50 MPa; flexural strength: 3 ~ 15 MPa) and energy absorption (toughness: 1 ~ 10 MPa·m[1/2]) have been developed[19,20]. Wang et al.[21] utilized 3D printing technology to develop a lightweight polymer-cement composite with hierarchical structure, which exhibited high toughness and low post-damage residual stress because the porous structure in macro scale enabled a periodic energy absorbance pattern during mechanical loading. Notably, compared with macro pores, micro and nano pores account for more than 95% of pores in cement-based materials, which plays a more important role in inhibiting crack initiation/propagation and therefore determining the strength and toughness of lightweight cement-based materials[22]. Nevertheless, the controlled design on the shape and size of micro and nano pores in cement-based materials has not been previously investigated within the relevant field.

In this research, a low-carbon emission, lightweight, strong and tough cement-based materials (LLST) was developed with hierarchically porous structure made up of micropores and nanopores, followed by the two steps as below. Firstly, the skeleton of LLST was constructed within 10 min due to the rapid gelation of hydrogel, which dominantly consisted of micropores. As cement hydrates desorpted on the surface of hydrogel skeleton, those micropores were gradually filled by newly-formed cement hydrates, which not only contained nanopores themselves but also were likely to turn the existing micropores to nanopores via filler effect. Additionally, compared with normal cement paste, LLST exhibited 54% reduction in density, 145% and 1460% improvement in specific compressive strength and fracture energy, respectively. Besides that, LLST displayed a more convenient fabrication process in 10 min and 20% higher utilization rate of raw materials. The current study not only provides valuable insights into the underlying mechanism of the advanced building materials with consolidation of lightweight, high strength and toughness, but also contributes to the realization of carbon neutrality in cement industry.

## Results and discussion
### Microstructure
The microstructure development over time in LLST-1 is shown in Fig. 1a–c. Microporous hydrogel skeleton with smooth surface and sponge-like structure was formed in 10 min in LLST-1 (Fig. 1a). Unhydrated cement particles were more likely to deposit on the hydrogel surface, driven by hydrogen bonding, ionic interactions, and van der Waals forces[23,24]. With cement hydration progressing over time, the sponge-like porous structure can be maintained in LLST at 7 days, as shown in Fig. 1b and S.1b, showing strong evidence of the template effect of hydrogel skeleton for cement hydration. Specifically, a large amount of water can be absorbed by hydrophilic groups in hydrogel

skeleton via hydrogen force, which promoted the in-situ dissolution of cement particles around the hydrogel skeleton. In that case, the cement hydrates tended to form around the network skeleton. Additionally, based on the heterogeneous nucleation theory[25,26], the nucleation barrier during the formation process of cement hydrates can be reduced by the nucleation sites from hydrogel skeleton. This phenomenon also facilitated the in-situ cement hydration on the hydrogel skeleton.

After 28 days of cement hydration, as shown in Fig. 1c and S.1c, it can be seen that part of micropores in hydrogel skeleton were filled by uniform dispersed cement hydrates containing nanopores, leading to the construction of micro/nano-hierarchical porous structure[27]. Normally, the pores in cement matrix are uncontrollable, varying randomly in size and shape. However, as shown in Fig. 1d and S.1c–f, it was observed that all the LLST showed sponge-like and hierarchical porous structures through a rapid gelation of hydrogel as skeleton and subsequent deposition of cement hydrates as skin in order. Compared with the irregular shape of pores with more tips in Reference samples (Ref.) (S.1c), the sponge-like hierarchical porous structure in LLST helps to mitigate the stress concentration and realize uniform stress distribution[28].

The spatial distribution of micropores in LLST-1 was examined using X-CT (Fig. 2a), where different pore sizes were labeled with different colors. It was clear to see that 1 ~ 50 μm micropores dominated in LLST-1, which was in sharp contrast to the 0.5 ~ 2.0 mm macropores present in foam cement[29]. In addition, such obtained three-dimensional uniform distribution of micropores in LLST-1 were expected to overcome the anisotropy mechanical properties of cement-based materials produced using ice-template method[10]. Nonetheless, a few pores larger than 50 μm were observed in LLST-1, attributing to the air bubbles introduced during the 10-min stirring, which suggested that the mixing parameters for hydrogel and cement mixture should be further optimized for a more precise control of micropores size in LLST.

The pore size distribution in LLST-(1-4) was measured using MIP (Fig. 2b, c) and NMR (S.2a). Compared with Ref., a significantly higher volume of 5 ~ 100 nm nanopores was present in all LLST. This was because the water absorption behavior of the hydrogel allowed more water to mix evenly with the cement particles, preventing the bleeding of water in the slurry due to density differences. In that case, the increased w/c ratio introduced more pores into the matrix, leading to a 12% ~ 54% reduction in the density of LLST compared to Ref. (S.2b). As observed, the current study provided a spontaneous structuring strategy to develop a hierarchical porous structure containing micropores (1 ~ 50 μm) and nanopores (5 ~ 100 nm) in cement matrix, which offered a promising approach for precisely controlling pore size distribution in cement-based materials.

### Mechanical properties
As shown in Fig. 3a, the flexural strength of LLST was higher than that of Ref. by 30% ~ 220%. Additionally, the midspan deformation rate and fracture energy of LLST, two typical toughness parameters, were 91% ~ 687% and 102% ~ 1365% higher than that of Ref. (S.3a). This enhancement in flexural strength and toughness was attributed to the hierarchical porous structures within LLST, as discussed in earlier section, which contributed to the higher fracture energy absorption, more uniform stress dispersion and better crack inhibition under mechanical load[30,31].

In addition, the specific compressive strength of LLST samples was 37% ~ 145% higher than that of Ref. and 120% ~ 1300% higher than that of conventional foam cement (Fig. 3b, c). This was because, compared with Ref. and foam cement, the sponge-like pores with hierarchical structure in LLST contributed to a more uniform stress dispersion and prevented the initiation and propagation of cracks during mechanical loading[28,32,33]. Therefore, a consolidation of

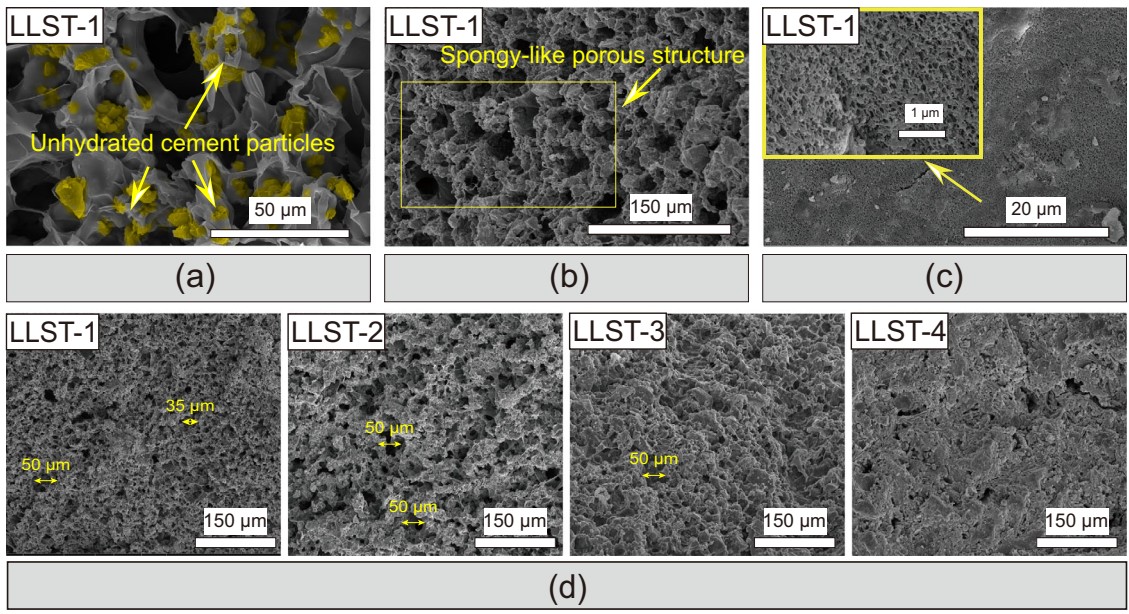

**Fig. 1 | Microstructure of LLST. a** The microstructure of LLST after 10 min of fabrication. **b** The microstructure of LLST after 7 days of curing. **c, d** The microstructure of LLST after 28 days of curing. LLST-1, LLST-2, LLST-3, and LLST-4 corresponded to LLST with water to cement ratio (w/c) of 2, 1, 0.67, and 0.5, respectively.

lightweight, strong and tough cement-based materials was reported in this study, with a density of 0.75 g/cm$^3$, specific compressive strength of 344 (kN/m$^2$)/(kg/m$^3$), flexural strength of 16.5 MPa, and fracture energy of 4865 KJ/m$^3$.

As observed from the compressive strain-stress curves of Ref. and LLST (S. 3b, c), Ref. demonstrated a typical quasi-brittleness failure mode in which a sudden drop in stress appears after the yield point[34]. In contrast, LLST exhibited a plastic failure mode with three periods, including the elastic regime, stress plateau and high densification[35] (Fig. 3d). The latter might be due to the deformation of the LLST pores, which consumed fracture energy during stress loading (Fig. 3e). More importantly, a strain-hardening phenomenon was found in cement-based materials without fiber reinforcement, suggesting that LLST possessed a higher failure threshold and better structural integrity. Specifically, LLST were composed of hierarchical porous structure (micropores and nanopores) and hierarchical bond strengths (covalent bonds in cement hydrates and hydrogel, ionic bonds and hydrogen bonds in cement hydrates/hydrogel skeleton, as illustrated in Fig. 3f), which enabled the multi-scale stress dispersion/mitigation capacity subjected to mechanical loading[36–38]. Consequently, the prior deformation of nanopores and preferential disruption of hydrogen bonds safeguard the integrity of micropores and the stability of covalent bonds, which was beneficial to raising the failure threshold.

**Cement hydration characterization**

Firstly, the XRD patterns in Fig. 4a and S.9a showed similar phase compositions in Ref. and LLST, including calcium hydroxide (CH), hydrated calcium silicate gels (C-S-H gels), monosulfoaluminate (AFm) and ettringite (AFt), indicating that the types of cement hydrates were independent on the presence of organic hydrogel. The addition of hydrogel, however, led to an increase in the peak intensity of AFt and AFm, which might be related to the diffraction preference orientation due to the large crystal size of AFt and AFm. SEM images revealed needle-like and hexagonal plate-like cement hydrates with large crystal sizes in LLST (Fig. 4b, c), resulting from that the sponge-like porous structure offered sufficient space for the growth of these cement hydrates. Notably, the needle-like cement hydrates (AFt/AFm) in larger sizes were expected to play similar roles as fiber in preventing the propagation of cracks and contributing to enhanced flexural strength in cement paste[39].

SEM images showed that the nucleation/growth sites of cement hydrates in LLST differed from Ref. For Ref., cement hydrates tended to grow on the surface of unhydrated cement particles[27]. For LLST, the cement hydrates mainly grew on hydrogel skeleton (Fig. 4c and S.9b). The growth pattern of cement hydrates in LLST can be explained as follows: on the one hand, the water-rich environment near the hydrogel skeleton facilitated the dissolution of cement particles and the formation of cement hydrates; on the other hand, the hydrogel skeleton played a role as hydration nucleation sites, which promoted the crystallization of cement hydrates on its surface. In summary, the dramatic change in crystal size and growth position of cement hydrates played a joint role in improving the mechanical strength of LLST.

The evolution of cement hydration heat was also impacted by the unique water supply mode in LLST. The heat flow of LLST, Ref. and hydrogel within 7 days were shown in Fig. 4e. Compared with Ref., LLST showed 10 ~ 20 h right shift in the occurrence of the acceleration period suggesting delayed cement hydration (Fig. 4e). This was because cement particles in LLST have restricted access to free water at the initial stage due to the rapid absorption of water by hydrogel (Poly(NIPAm-SA) was observed to absorb water 150 times of its weight[40]) inhibiting the early-age hydration. Nonetheless, this absorbed water can be slowly released for cement hydration over a prolonged time owing to the gradients in ionic concentration, the capillary force and the consumption by hydration reaction. Normally, the lower heat release of LLST benefits the internal temperature reduction of concrete in the early stages of practical engineering. Therefore, such a unique water supply mode in LLST was believed to positively control the cement hydration reaction and reduce the risk of early-age thermal cracking in mass concrete (dams, nuclear power plants)[41].

Specifically, hydration degree in LLST reached 77% ~ 87% (7 days) and 78% ~ 92% (28 days), which was 2% ~ 21% higher than that in Ref. (Fig. 4f). This observation was attributed to the unique role of hydrogel in LLST, serving as a uniform & controlled water supply source, as discussed in earlier sections. Normally, in conventional cement-based materials, the direct mixing method often leads to a portion of water being unable to participate in cement hydration, due to water evaporation and capillary adsorption between cement particles[42]. However, in the current study, water evaporation and capillary adsorption can be fixed through the water sustained-release mode driven by hydrogel water desorption, which helped to improve the utilization

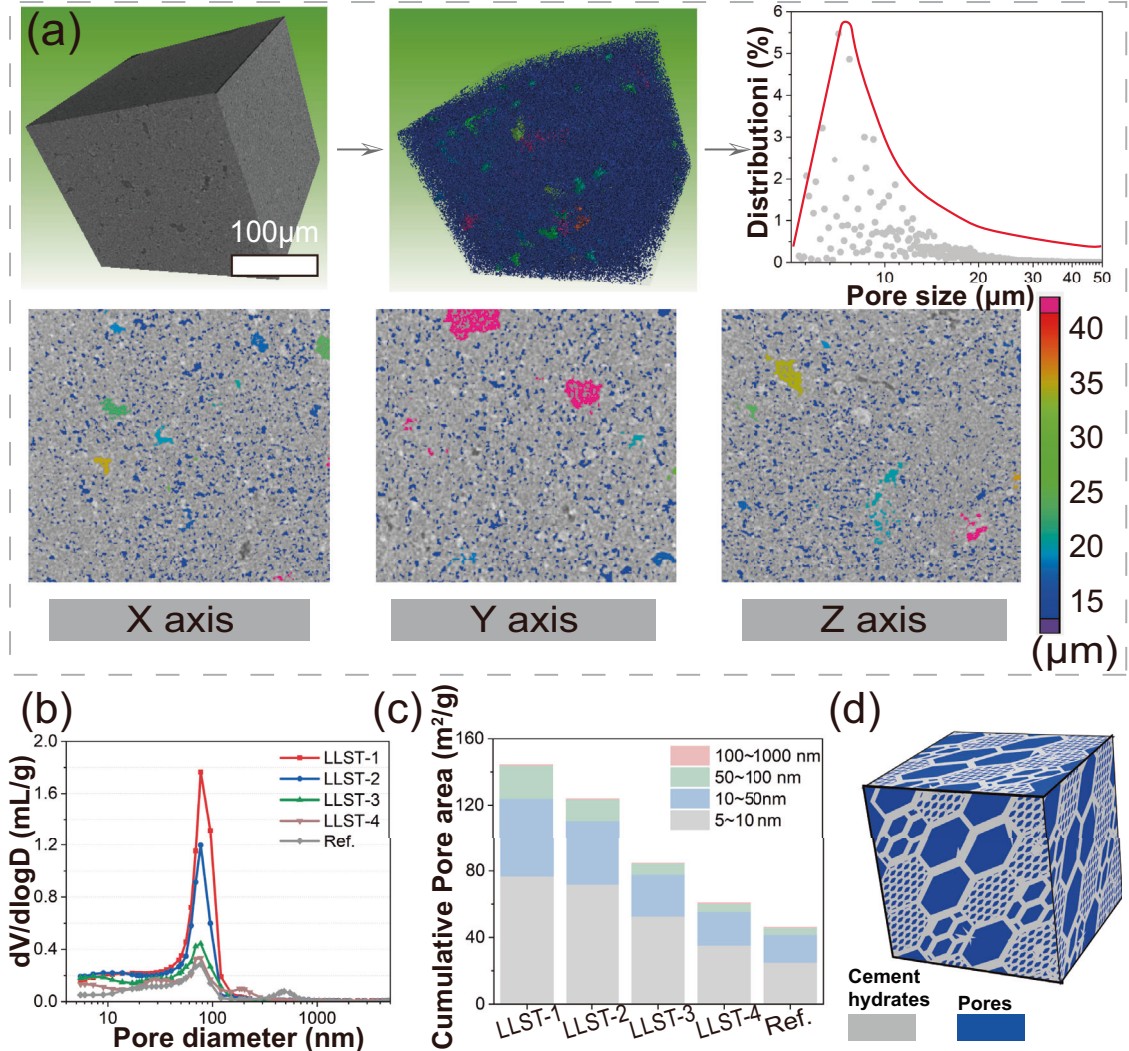

**Fig. 2 | Pore structure of LLST. a** Spatial distribution of micropores in LLST-1 obtained by X-ray Computed Tomography (X-CT), where different pore sizes were labeled with different colors. **b**, **c** Size distribution and cumulative volume of nanopores detected by Mercury intrusion porosimetry (MIP). **d** Schematic hierarchical micro/nanoporous structure. LLST-1, LLST-2, LLST-3, and LLST-4 corresponded to LLST with w/c of 2, 1, 0.67, and 0.5, respectively. Ref. represented pure cement with a w/c ratio of 0.5.

rate of raw materials produced for cement-based materials. In conclusion, the revolutionized water supply mode of cement hydration, changing from directly mixing to water sustained-release by hydrogel, effectively addressed the challenge of rapid reaction in the early stage and insufficient reaction in the later stage. Notably, there was a discrepancy between the sample size in this study and those encountered in practical engineering applications, which should be addressed in further research.

Furthermore, FT-IR was carried out to reveal the interaction between the hydrogel and cement hydrates, as shown in Fig. 4. Firstly, the peaks at 3650 cm⁻¹, 3437 cm⁻¹, and 1650 cm⁻¹ in Ref., as displayed in Fig. 4g, were ascribed to the vibrations of the O-H, while the peak at 975 cm⁻¹ was related to the vibrations of Si-O. In comparison, FT-IR spectra of LLST exhibited additional peaks related to the hydrogel[43] associated with the vibrations of O-H (3650 cm⁻¹, 3437 cm⁻¹), C = O (1650 cm⁻¹, 1429 cm⁻¹) and N = O (1116 cm⁻¹, 876 cm⁻¹), as shown in Fig. 4h, i. Notably, the O-H peaks at 3650 cm⁻¹ and 3437 cm⁻¹ in LLST become wider than that in Ref., indicating that a stronger hydrogen bonding was generated in LLST due to the interaction between Ca and Al ions released from cement hydration with -COO⁻ in hydrogel[44], which was beneficial for the compressive enhancement due to the better interfacial compatibility.

## Mechanisms of interactions between the Ca ion and hydrogel

The interactions between hydrogel and cement particles were further studied by machine learning force field molecular dynamics (MLMD) along with well-tempered metadynamics (WT-MetaD) simulations. First, we explore the most stable state of the interaction between the Ca ion and the carboxyl group of the hydrogel by employing the coordination number of the Ca ion with the O ion from the water ($CN(Ca-O_w)$) and the coordination number of the Ca ion with the O ion from the carboxyl group ($CN(Ca-O_c)$) as reaction coordinates. Figure 5a shows the free energy surface (FES) of the ligand exchange of the Ca ion by water and carboxyl group. The most stable state on the FES is located at the state A with reaction coordinate of (5, 1). The structure of the state A (Fig. 5d) presents that, upon interaction with the carboxyl group, the Ca ion forms a single Ca-O bond with the carboxyl group, while forming five Ca-O bonds with surrounding water molecules. Other stable states (shown in red on the FES) indicate that the Ca²⁺ ion maintains a coordination number between five and seven during the interaction, which is consistent with experimental findings[45].

Further, we investigated the reaction pathways for the approach of the Ca ion to the hydrogel to identify the intermediate structures and the free energy barriers. The distance between the Ca ion and one

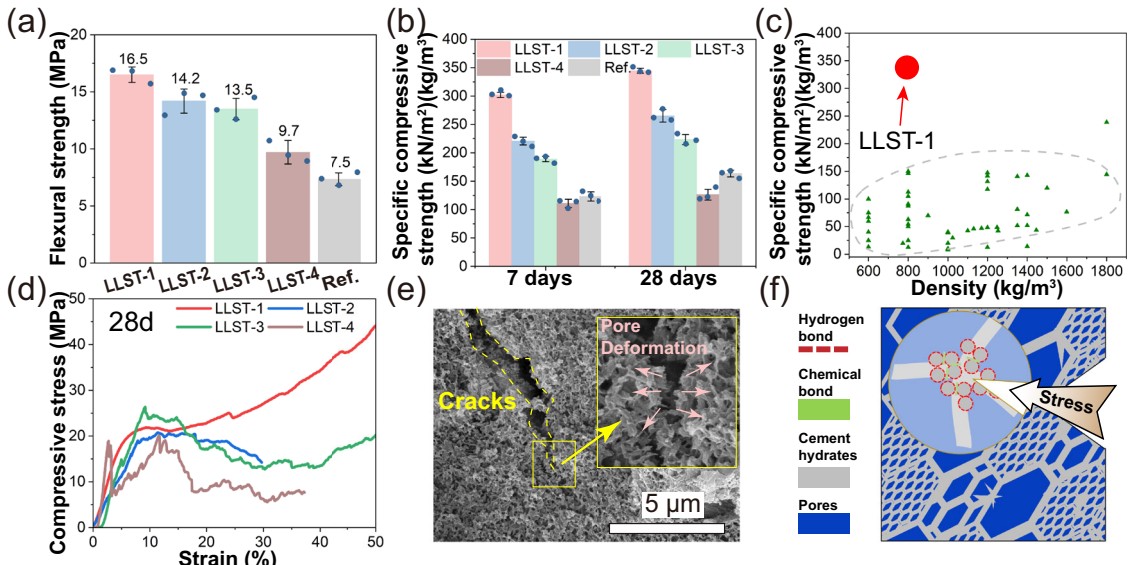

**Fig. 3 | Mechanical properties of LLST. a** Flexural strength of LLST after 28 days of curing. The error bars in Fig. 3. **a, b** represented the standard deviation, calculated based on three replicate experiments. **b** Specific compressive strength and **c** comparison with reported foam cement[9,32]. **d** Strain-compressive stress curve of LLST. **e** Crack propagation was inhibited by pore deformation. **f** Schematic diagram of multi-scale stress dispersion.

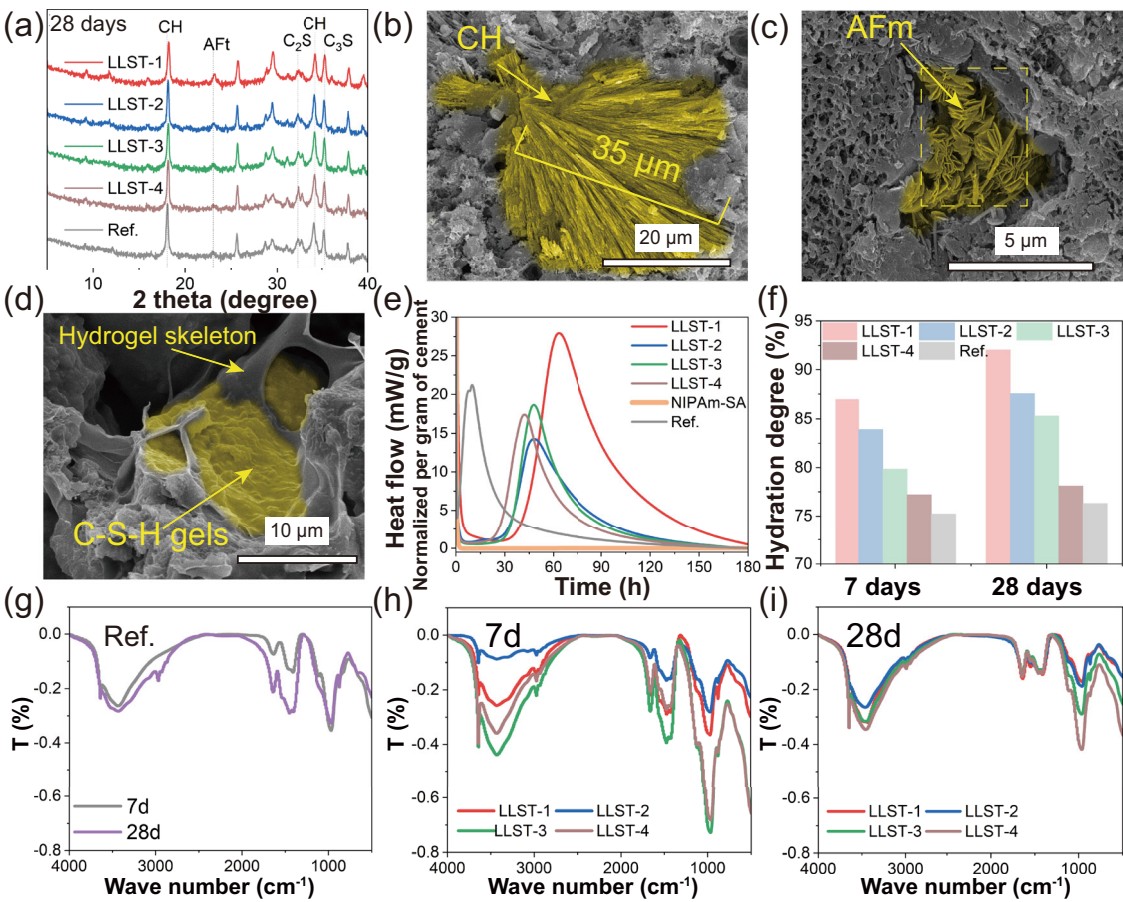

**Fig. 4 | Hydration characteristics of LLST. a** X-ray diffraction (XRD) pattern of cement hydrates in LLST and Ref. after 28 days of curing. CH, $C_2S$, $C_3S$, AFt, AFm corresponded to calcium hydroxide, dicalcium silicate, tricalcium silicate, ettringite and monosulfoaluminate, respectively. **b–d** Morphology of cement hydrates after 28 days of curing. **e** Hydration heat evolution within 7 days curing. **f** Hydration degree of LLST. Each sample was tested once. **g–i** Fourier Transform Infrared Spectrometer (FT-IR) spectra of Ref. and LLST after 7 and 28 days of curing.

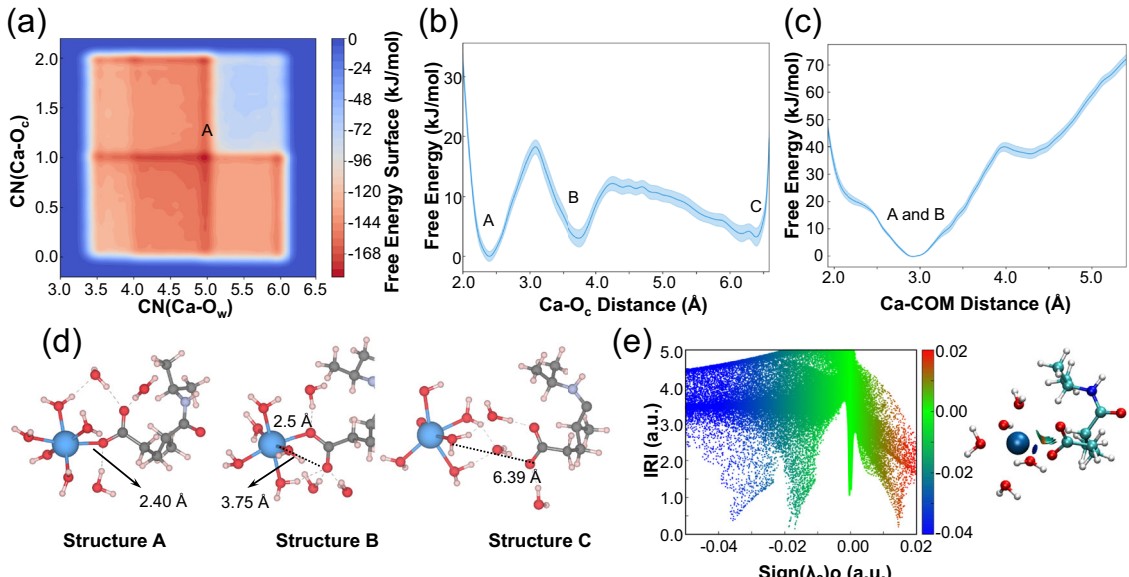

**Fig. 5 | Molecular interaction mechanisms between the carboxyl group from hydrogel and the Ca ion released by cement particles. a** Free energy profile with respect to the coordination number of the Ca ion with the O ion from the water (CN(Ca-O$_w$)) and the coordination number of the Ca ion with the O ion from the carboxyl group (CN(Ca-O$_c$)). **b** Free energy profile with respect to the distance between the Ca ion and one of the O$_c$ ions. **c** Free energy profile with respect to the distance between the Ca ion and center of mass (COM) of two O$_c$ ions. The shaded regions in (**b** and **c**) show the errors calculated by block analyses. **d** Configurations of the free energy minimum states on the FES. The blue, black, light purple, red, and white spheres are indicted to the calcium, carbon, nitrogen, oxygen and hydrogen. **e** Scatter map between sign($\lambda_2$)ρ and IRI and isosurface map of IRI. Indigo, cyan, blue, red, and white spheres are indicted to the calcium, carbon, nitrogen, oxygen and hydrogen.

of the O$_c$ ions was chosen as the reaction coordinate. Figure 5.b shows that there are three local minima on the FES. State C, with a Ca-O$_c$ distance of approximately 6.39 Å, represents a scenario where there is no interaction between the Ca$^{2+}$ ion and the hydrogel (Fig. 5d), where the Ca ion is isolated from the hydrogel by water molecules via hydrogen bonds. Upon crossing a free energy barrier of approximately 8.95 kJ/mol, the Ca ion approaches the hydrogel, forming a Ca-O bond that is more stable than the isolated state. States A and B, with Ca-O$_c$ distance of 2.40 Å and 3.75 Å, represent two stable states when the Ca ion interacts with the carboxyl group of the hydrogel. The molecular structures of states A and B (Fig. 5d) reveal that both configurations involve the formation of a single Ca-O bond with the carboxyl group. This result aligns with the results from the FES with CN as reaction coordinates. In state B, although the Ca ion does not coordinate with the biased O$_c$ ion, it forms bond with another O$_c$ ion with bond length of 2.5 Å. The most stable state on the FES is the state A, in which the Ca-O$_c$ bond is 2.40 Å. Compared with the state B, the lower free energy of state A is due to the greater electronegativity of the O$_c$ ion bonded with the Ca ion. This asymmetry of the charge distribution between two O$_c$ ions arises from the differing interaction strengths between water molecules and the carboxyl in solutions. Additionally, we conducted a simulation where the reaction coordinate was the distance between the Ca ion and the center of mass (COM) of the two O$_c$ ions to validate the reaction product of the Ca-hydrogel interaction. The FES shows that the most stable state occurs at a distance of ~2.91 Å, which corresponds to states A and B in the FES with reaction coordinate of Ca-O$_c$ distance, and state A in the FES with the reaction coordinates of CN(Ca-O$_w$) and CN(Ca-O$_c$). To characterize the nature of the Ca-hydrogel interaction, we performed an interaction region analysis (IRI) analysis (Fig. 5e), a novel real-space function that identifies both weak interaction and chemical bond. The IRI analysis reveals a blue region between the Ca ion and one of the O$_c$ ions, indicating a strong electrostatic attraction and the formation of a chemical bond. An orange region occurs between the Ca ion and center of carboxyl group suggests a steric hindrance effect. These findings are consistent with our experimental observations.

## Discussion

In this research, nature-inspired lightweight building materials with low carbon emission, high strength and ultra toughness was developed by tailoring hierarchically porous structure, using spontaneous structuring strategy. The intention of using hydrogel is to fully utilize the uniform micro pores in hydrogel as a template to construct cement paste with hierarchical porous structure, which was accomplished by a rapid gelation of hydrogel as skeleton and subsequent deposition of cement hydrates as skin in order. In this case, the pore shape turns to be more regular with sponge-like structure, and pore size is hierarchically in the range of micropores (1 ~ 50 μm) and nanopores (5 ~ 100 nm) in LLST, which contributes to the lightweight, strong and tough design of cement-based materials. In addition, as compared with normal cement paste, LLST displayed 54% and 49% reduction in density and carbon emission, 145% and 1365% improvement in specific compressive strength and fracture energy, and such a significant improvement was attributed to the multi-scale stress dispersion from the hierarchical porous structure and multi-strength of bonds in LLST. Furthermore, the transformation of water supply mode, from the free water in the space of matrix (conventional cement-based materials) to the confined water on skeleton of hydrogel (the newly developed LLST), contributed to a more homogeneous structure and better interfacial compatibility between the hydrogel skeleton and cement hydrates in LLST, and thus led to the enhancement of strength and toughness of cement-based materials. The interactions in hydrogel-Ca system are mainly due to the strong chemical bonding between the Ca ion and the O ion from the carboxyl according to machine learning based atomistic simulations, which is consistent with the current experimental results.

## Methods
### Materials

**Raw materials.** The analytical grade N-isopropylacrylamide (NIPAm), sodium acrylate (SA), N,N-methylenebisacrylamide (Bis), potassium persulfate (APS), and N,N,N'N'-tetramethylethylenediamine (TEMED) were purchased from Shanghai Macklin Biochemical Co., Ltd. Cement

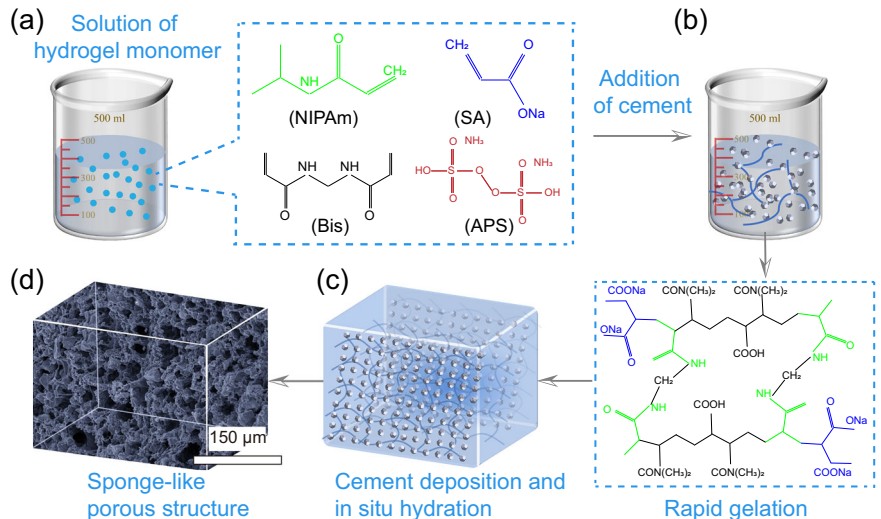

**Fig. 6 | The fabrication process of LLST. a** The hydrogel ingredients (NIPAm, SA, Bis, and APS) were dissolved in DI water. **b** Cement particles were then added to the solution. **c** After the gelation of hydrogel, cement particles were deposited on its surface. **d** Following cement hydration, LLST exhibited a sponge-like porous structure.

(P·I 42.5 type) was supplied by China Building Materials Academy. The composition of cement was shown as S.Tables 8 and 9 in supplementary files. Deionized water (DI) was used in this research.

**Fabrication process of LLST.** The mixing process of LLST samples are shown in Fig. 6. Firstly, the ingredients of hydrogel (NIPAm, SA, Bis and APS) were sequentially dissolved in DI water, and the cement particles and TEMED were added within the above solution to initiate the cross-linking reaction. Continuous stirring was maintained during the mixing process to ensure the uniform dispersion of cement particles without sedimentation. During which the initiator was released by monomer, and then combined with each other to form the polymer chains. Subsequently, polymer chains were interconnected by cross-linking agent to form a polymer network. The above gelation process of hydrogel was completed within 10 min as shown in Fig. 7. LLST has undergone a solution-to-solid transformation and exhibited hydrogel-specific elastic behavior. After gelation, the mixture was transferred into a curing container with a temperature of 20 °C and RH of 95% for cement hydration until test. The pore structure of LLST was primarily affected by the concentration of ions in the solution and the presence of unhydrated cement particles. Therefore, four mix designs of varying dosages of cement were studied for controlled pore structure in LLST, as detailed in Table 1. The pure cement sample was named as Reference (Ref.).

**Characterization**
**Microstructure and functional group.** The microstructure of LLST was examined by SEM (Navo Nano SEM450, Thermo Fisher Scientific, USA) with 3 spot sizes and 15 kV acceleration voltage. Before the test, the mixture was placed in a curing container with a temperature of 20 °C and RH of 95%. At testing ages (after 7 and 28 days of cement hydration), LLST was frozen in liquid nitrogen for 10 min before being dried in a lyophilizer below −50 °C for 8 h. After that, LLST was fractured and sputter-coated with gold for SEM analysis. In addition, the chemical functional groups of LLST (powder, <75 μm) were analyzed using FT-IR from (Madison-WI, Nicolet, USA).

**Mechanical strength.** Specimens after 7 and 28 days of curing (20 mm cube for compressive test and 20 × 20 × 80 mm cuboid for flexural test) were dried in a forced ventilation oven at 40 °C for 24 h to remove water. Before the test, the load surfaces of specimens for compressive strength test were caped. Specimens were then tested using a

Mechanical Universal Testing Machine (UTM5105, SUNS, China) for compressive strength (loading rate 5 mm/min) and flexural strength (loading rate 1 mm/min) following standard ASTM C513. The fracture toughness of LLST was evaluated following the recommendation proposed draft by RILEM Committee[46,47]. The specimens (20 × 20 × 80 mm) were cut at mid-span with a half depth, and the loading rate was set as 0.02 mm/min (Mechanical Universal Testing Machine, UTM-6503, SUNS, China). The fracture toughness (KIC) of LLST was calculated according to Eq. 1. Three specimens of each mix design were tested to obtain an average mechanical strength value with standard deviation.

$$K_{IC} = \frac{P_Q S}{BW^{3/2}}\left[2.9\left(\frac{a}{W}\right)^{1/2} - 4.6\left(\frac{a}{W}\right)^{3/2} + 21.8\left(\frac{a}{W}\right)^{5/2} - 37.6\left(\frac{a}{W}\right)^{7/2} + 38.7\left(\frac{a}{W}\right)^{9/2}\right]$$

(1)

where $P_Q$ represented the peak load from the load-displacement curve; $S$, $B$, and $W$ was related to the span, width and height of the specimen; $a$ represented to the notch depth.

**Pore structure**
The pore structure of LLST after 28 days of curing, characterized by an approximately cylindrical shape (height and diameter: 1 - 2 mm), was analyzed using three techniques. The spatial pore distribution was characterized by X-CT (Zeiss Xradia 510 Versa, Germany), equipped with a high-resolution detector (2000 × 2000 pixels, 120 kV accelerating voltage and 80 μA beam current). The pore distribution was measured using MIP (Autopore IV9510, Micromeritics, USA). After 28 days of curing, the samples for MIP test were frozen in liquid nitrogen for 10 min, and then dried in a lyophilizer below −50 °C for 24 h. After that, LLST was split into small pieces around 2 mm. 1H Nuclear Magnetic Resonance Spectroscopy (NMR) (MesoMR12-060V-I, Niumag, China) was also used in this research to detect the porosity of of specimens. Before the NMR test, LLST specimens (20 mm cube) was put in a vacuum saturated device for 24 h to achieve water-saturated state, which allowed for the estimation of porosity by inverting the relaxation time of water in various pore sizes. The transverse relaxation time (T2) distribution was measured under constant magnetic field of 0.42 T and 18 MHz, echo spacing of 110 μs, echo numbers of 500, and 200 ms, during the whole test. Each sample for X-CT, MIP and NMR tests was tested once.

**Hydration kinetics.** Quantitative X-ray diffraction test was performed on an AXS D8 ADVANCE (Bruker, Germany) diffractometer using CuKα radiation at 40 kV and 40 mA, $2\theta = 5°–70°$, and a step width of $0.02°•s^{-1}$ to evaluate the mineralogical phases of LLST specimens after 7 and 28 days of curing. The internal standard corundum was adopted with full Rietveld analysis via Topas3-C software. The residual weighted profile value in all specimens was in the range of 7 - 10. The peak intensity results were obtained after plotting log scale on Y-axis. Cement hydration degree was calculated based on the content of clinker phase ($C_3S$, $C_2S$, $C_4AF$, and $C_3A$) obtained in QXRD[48]. Three samples were tested for each group to obtain the average results. In addition, TAM Air isothermal calorimeter (TA Instruments, USA) was used to measure the heat of hydration for 7 days at 25 °C. To maintain a consistent temperature, all raw materials were stored in a thermostatic chamber set to 20 °C for 24 h before the test. The heat evolution of LLST, Ref. and hydrogel within 7 days were recorded, with each sample tested once.

## Computational methods
**Model construction.** The porous solid phase of hydrogel in LLH cement is a copolymer system, which is composed of NIPAM and SA monomers as well as Bis molecules (S. 6). During the process of in-situ polymerization of monomers with cement hydration, a copolymer skeleton is firstly constructed, followed by the coordination of the mental ions (mainly $Ca^{2+}$) released by cement particles by carboxyl group from hydrogel[49]. Thus, it is more likely that the site of interactions between hydrogel and the ions released by cement particles is in the pore solution environment. Based on this promise, the model should be constructed by putting polymer in solutions containing water molecules, $Ca^{2+}$, and $SiO_4^{2-}$ ions. Since the interaction between

the Ca ion and the carboxyl group is the primary reaction between the hydrogel and aqueous species in the cement pore solution, we isolate the local structure of the hydrogel composed of one NIPAM and one SA unit (with the Na ion removed) for this model. The NIPAM-SA monomer, together with a single Ca ion, is placed within an explicit solvation shell with a hydration radius of 5.5 Å (the density of water is $1 g/cm^3$). This simplified model is intended to generate a training dataset for the machine learning force field (MLFF) using *ab* initio molecular dynamics (AIMD) simulations. The final model consists of 481 atoms within an orthogonal simulation box with dimensions of $16.6 Å × 17.2 Å × 17.6 Å$.

**Dataset preparation.** The training dataset was constructed by AIMD coupled with well-tempered metadynamics (WT-MetaD) simulations. AIMD was conducted using the freely available CP2K software[50]. The Goedecker-Teter-Hutter (GTH) pseudopotentials were used to model the core electron effects[51,52]. While the Gaussian type of double-ζ valence polarized (DZVP) basis set[53] was used to describe the valence electrons. Long-range electrostatic forces were computed using an additional plane wave basis set with an energy cutoff of 400 Ry. The generalized gradient approximation with the Perdew-Burke-Ernzerhof (PBE) functional[54] was chosen to describe the exchange-correlation effects. The Grimme D3 approach[55] was used to correct the total energy, energy gradient and frequencies due to the London-dispersion interactions in the system. The Brillouin zone was sampled at the Γ-point due to the large size of the simulation cell. The hydrogen was replaced by deuterium to increase the integration time for each step to 1 fs[56,57]. The canonical sampling through velocity rescaling thermostat[58] with a time constant of 200 fs was coupled to the system to keep temperature at 300 K in the canonical ensemble (NVT). The convergence criterion for wave function optimization was set by an energy difference tolerance of $10^{-12}$ Hartree and self-consistent field cycles of $10^{-6}$ Hartree. The system was first optimized to a stable state and then thermalized for at least 5 ps prior to the metadynamics simulation.

WT-MetaD setup was implemented in the PLUMED code[59,60]. The collective variable for the generation of the training dataset was chosen as the distance between the Ca ion and the center of mass of two O ions from carboxyl group. To confine the exploration within reasonable bounds, two quadratic walls were introduced with a force constant of 500 kJ/mol, and their positions are listed in Table 2. Besides, the initial height, width, and deposition frequency of Gaussian hills, as well as the biasfactor are also shown in Table 2.

The AIMD-based WT-MetaD were run for 100 ps, generating a total of 100,000 configuration frames. From this trajectory, approximately 5000 structures were randomly selected to form the training dataset, while 200 structures were chosen for the validation dataset. To capture system behavior at higher energy levels on the potential energy surface, an additional set of 150 structures was randomly selected, with perturbations of 5%, 10%, 15%, 20%, and 30% (30 structures for each perturbation level). These perturbed structures were added in the training dataset after performing DFT calculations with the same level of accuracy as the AIMD simulations.

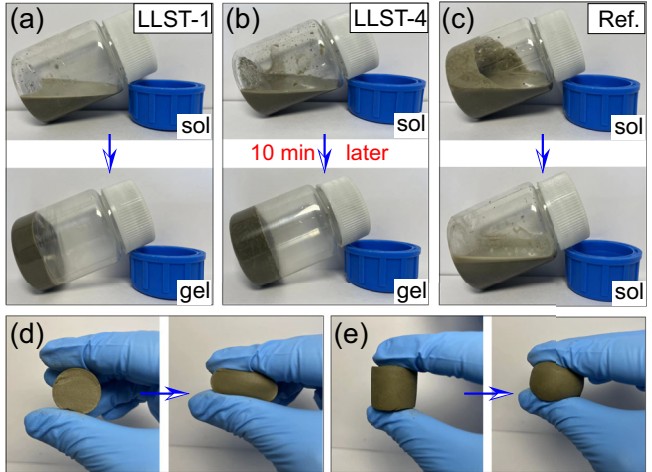

**Fig. 7 | Gelation and elastic behavior of LLST. a–c** After 10 min of mixing, LLST-1 and LLST-4 transitioned from a liquid to a solid state, whereas the Ref. group remained in the liquid state. **d, e** The elastic behavior of LLST after gelation.

### Table 1 | Mix designs of different LLST

| Mix. (g) | LLST-1 | LLST-2 | LLST-3 | LLST-4 | Reference (Ref.) | NIPAm-SA |
|---|---|---|---|---|---|---|
| Cement | 20 | 40 | 60 | 80 | 80 | – |
| Water | 40 | 40 | 40 | 40 | 40 | 40 |
| NIPAm | 3.2 | 3.2 | 3.2 | 3.2 | – | 3.2 |
| SA | 0.8 | 0.8 | 0.8 | 0.8 | – | 0.8 |
| Bis | 0.06 | 0.06 | 0.06 | 0.06 | – | 0.06 |
| APS | 0.06 | 0.06 | 0.06 | 0.06 | – | 0.06 |
| TEMED (µL) | 60 | 60 | 60 | 60 | – | 60 |

**Table 2 | Parameters of WT-MetaD simulations**

| Project | CV | Quadratic walls | | Initial height | Width | Deposition frequency | Biasfactor |
|---|---|---|---|---|---|---|---|
| | | Lower position | Upper position | | | | |
| AIMD | Ca-COM Distance | 1.8 Å | 6.8 Å | 3.5 kJ/mol | 0.1 Å | 30 fs | 20 |
| MLFF 1 | CN(Ca-O$_c$) | 3.5 | | | 0.1 | | |
| | CN(Ca-O$_w$) | 0.1 | | | 0.05 | | |
| MLFF 2 | Ca-O$_c$ Distance | 2 Å | 6.5 Å | | 0.1 Å | | |
| MLFF 3 | Ca-COM Distance | 2 Å | 5.5 Å | | 0.07 Å | | |

**Training machine learning force field.** The machine learning force field (MLFF) was trained using the DeepMD-kit package (version of 3.0.0b4)[61,62]. We fine-tuned the DPA-2 model based using our dataset to enhance the accuracy of the MLFF. DPA-2 is an advanced large atomic model architecture, pre-trained on a diverse set of chemical and materials systems via a multi-task approach. This reduces the amount of data required for fine-tuning and improves sample efficiency[63]. The three-body embedding approach, DeepPot-SE, was used to represent the potential energy surface. A cut-off radius of 6.0 Å was adopted to represent the local internal structure around any atom. To mitigate the discontinuity caused by this cut-off, a cosine-smoothing function was applied starting at a distance of 0.5 Å. Similarly, the start smoothing and cut-off radius in the three-body representation was set to 0.5 and 4.0 Å, respectively. The embedding and fitting networks sizes were configured as (25, 50, 100) and (240, 240, 240), respectively. The learning rate exponentially decays from $1.0 \times 10^{-3}$ to $3.51 \times 10^{-8}$ every 5000 training steps. The batch size was automatically adjusted during the training process. The perfectors of the energy and force in the loss function were dynamically updated, ranging from 0.02 to 1 for energy, and from 1000 to 1 for force, over the course of the optimization. The MLFF was trained for $1.0 \times 10^5$ steps.

**Validation of the machine learning force field.** The accuracy of the MLFF was validated by comparing the energies and atomic forces calculated by the MLFF with those obtained from DFT calculations for both the training and testing datasets. We randomly selected 200 configurations from the initial AIMD-based WT-MetaD simulations, 150 configurations from the perturbed structures and 200 configurations form testing dataset. The overall root mean squared errors were $7.4 \times 10^{-4}$ eV/atom for energy and $8.4 \times 10^{-2}$ eV/Å for atomic force. Correlation plots (Fig. S4) reveal a strong linear relationship between the energies and atomic forces predicted by the MLFF and those computed using DFT, indicating the reliability of the MLFF for both datasets.

**Machine learning force field molecular dynamics simulations.** All MLMD simulations were performed in the LAMMPS code[64] using the fine-tuned model. The periodic boundary conditions were applied in all directions. The timestep of 1.0 fs was adopted. The initial config-uration was first performed energy minimization with conjugate gradient algorithm and then equilibrated at the chosen temperature with NPT ensemble for 100 ps. In production run, the NVT ensemble was applied, and the temperature was controlled by the Nose-Hoover thermostat with a relaxation time of 100 fs. The duration of the production run was 1 ns for each simulation.

To comprehensively elucidate the molecular mechanisms underlying the interaction between the Ca ion and the carboxyl group, three collective variables were chosen for the WT-MetaD simulations. In the simulation 1 (MLFF 1), we aimed to determine the most stable configuration for the binding of the Ca ion to the car-boxyl group. To achieve this, the coordination numbers (CN) of the Ca ion with the O ion from water molecules (O$_w$) along with the CN of the Ca ion with the O ion from carboxyl group (O$_c$) were adopted to

form the two-dimensional CVs. The CN was computed using the following equation (Eq. 2):

$$CN\left(Ca - O_{w/c}\right) = \sum_{j \in O_{w/c}} s_{ij}\left(r_{ij}\right) = \sum_{j \in O_{w/c}} \frac{1 - \left(\frac{r_{ij} - d_0}{r_0}\right)^n}{1 - \left(\frac{r_{ij} - d_0}{r_0}\right)^m} \quad (2)$$

where $r_{ij}$ is the distance between atom i and atom j. $s_{ij}\left(r_{ij}\right)$ is a switching function describing the coordination between atom i and j. $d_0$ is the central value of the function. $r_0$ is the acceptance distance of the switching function. The $d_0$ and $r_0$ were set as 2.42 Å and 0.4 Å, respectively, which were tested to be appropriate[65,66]. n and m are 6 and 12, respectively[67,68].

In simulation 2 (MLFF 2), the distance between the Ca ion and one of the O$_c$ ions was chosen to explore the reaction mechanism as the Ca ion approaches the carboxyl group. In simulation 3 (MLFF 3), the distance between the Ca ion and the center of mass (COM) of two O$_c$ ions was selected to verify the rationality of the bonding mechanism found by different choice of CVs. The parameters of the MLFF-based WT-MetaD are shown in Table 2. Convergence tests for free energy surfaces, time evolution of the CVs, time evolution of the height of bias are shown in Figs. S.5–7. The errors in the free energies for MLFF 2 and MLFF 3 were calculated using the block-analysis method[69]. The average error converged at the block size of around 900 in both simulations (Fig. S.8). Using 900 blocks, the average errors were calculated to be 1.22 kJ/mol for MLFF 2 and 1.42 kJ/mol for MLFF 3.

**Wavefunction analyses.** Interaction region indicator (IRI) were per-formed using the Multiwfn software[70]. All wavefunction analyses were based on the wavefunctions calculated at the B3LYP/6-311 G* level with implicit solvent model (water).

## Data availability
The data generated in this study are provided in the Source Data file. The datasets and models used in this study, as detailed in the sup-plementary materials, are all available on AIS Square (https://www.aissquare.com/models/detail?pageType=models&name=DPA-2-Ca-NIPAM-SA&id=284). Atomic coordinates of the optimized computa-tional models and trajectories of machine learning molecular dynam-ics is available in the figshare: https://doi.org/10.6084/m9.figshare.28498718.v1. Source data are provided with this paper.

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

## Acknowledgements

The authors would like to acknowledge the financial supports from National Key Research and Development Program of China (Grant No. 2023YFE0205400), National Science Foundation for Distinguished Young Scholars (Grant No. 51925903), National Natural Science Foundation of China (Grant No. 52378224), Funds for International Cooperation and Exchange of the National Natural Science Foundation of China (Grant No. 52261160646) and the Macao Science and Technology Development Fund (FDCT) (Grants No. 0074/2023/RIB3 and 0139/2024/RIB2), and the Macau University of Science and Technology Faculty Research Grant (Grant No. FRG-24-085-FIE). The authors also acknowledge Beijing PARATERA Tech CO., Ltd. for providing HPC resources to perform the simulations.

## Author contributions

Jinyang Jiang: Conceptualization, Methodology, Investigation, Resources, Writing-review & editing, Supervision. Han Wang: Conceptualization, Methodology, Data curation, Writing-original draft, Writing-review & editing. Junlin Lin: Conceptualization, Writing-original draft, Writing-review & editing. Fengjuan Wang: Methodology, Writing-review & editing. Zhiyong Liu: Methodology, Writing-review & editing. Liguo Wang: Conceptualization, Formal analysis, Writing-review & editing. Zongjin Li: Conceptualization, Writing-review & editing. Yali Li: Conceptualization, Investigation, Writing-review & editing, Data curation. Yunjian Li: Investigation, Methodology, Formal analysis, Writing-original draft, Writing-review & editing, Supervision. Zeyu Lu: Resources, Conceptualization, Investigation, Writing-review & editing, Data curation, Funding acquisition, Supervision.

## Competing interests

The authors declare no competing interests.
