## [Transparent Peer Review file · Nature Communications]

Nature-Inspired Hierarchical Building Materials with Low CO₂ Emission and Superior Performance

Corresponding Author: Professor Zeyu Lu

Version 0:

Reviewer comments:

Reviewer #1

(Remarks to the Author)

Dear Authors, thanks for the manuscript, which was submitted to Nature Communications.

I have thoroughly read your paper, however, I do not agree on the novelty and importance of the findings.

You are describing an in-situ-polymerization of polyacrylamide with crosslinkers and acrylic acid, forming to a polymer gel, which works as a template for cement hydration. Such copolymerizations have been observed in the past, the most important works were done by Cölfen.

You describe a higher tensile strength. Looking at the pictures and data, I would expect that this is some kind of fiber-reinforced structure, which provides similar performance. You also show performance on compressive strength, comparing with foam cement. Air is important for the strength, air size as well. Unfortunately, there is no information about the reference foam cement, where you state a higher strength value.

You describe a nucleating effect of acrylic acids. This is akin to the findings of FLatt, who showed that highly charged PCEs show a nucleating effect on Ettringite in the pore solution.

The effect of water storage is well known for superadsorber, which are also composed of Polyacrylamides.

Additionally to the general importance of the paper, I have some concerns about the technical quality of the paper.

1) The polymerization is not properly described and evaluated. APS does not start polymerization at room temperature. You have to heat, usually 70-100°C. You don't describe that. This impacts hydration of cement. Not polymer analytics have been done despite some pictures.

2) Calorimetry has not been described. I cannot see the reaction heat of the polymer. The hydration of LLST1 looks a bit high, having the least cement. For concluding on strength, you would need cumulative heat. Why does the reference cement have no sulfate depletion peak?

3) Degree of hydration: On which basis has this been calculated? Average phase composition? C3S only?

4) Particle size of ettringite has a huge effect on tensile strength of cement. You have to consider in the discussion.

5) What is the chemistry of your cement?

6) I don't understand how you calculate CO₂. Please provide the underlying data and considerations you used.

On the modelling, I cannot give proper feedback.

Reviewer #2

(Remarks to the Author)

This is an interesting work on cementitious materials prepared through gelation, which provides astonishing performance results compared to materials of such type that are manufactured via traditional approaches. The work may be of interest for a broad readership. However, it presently appears characterized by a number of significant shortcomings that require a major revision for a new assessment. The following major points motivate the opinion of this reviewer on this work:

- The novelty of this work relative to the state-of-the-art is poorly justified.
- The study makes several statements that are poorly justified. This issue should be fixed, either by providing results or adequate references in support of such statements. Without this groundwork, parts of this study appear inadequately supported by data.
- The study is inherently limited by a problem of scale, as extremely small samples are prepared and tested. What is the applicability of this approach to larger volumes of materials. Without proven applicability, this work may remain only a theoretical exercise, as no structures made of LLST may be built in practice in the future.
- The experiments use DI water to prepare the considered samples. What is the influence of ionic species that are always present in the water used to prepare cement-based materials on the results of this work?
- Several testing features and modeling assumptions must be provided, as they have been presently omitted.
- The English language should be revised in detail by a native speaker, as many many grammatical errors arise from the start of the abstract (e.g., L. 18 is unclear; L. 23 presents an error; etc.). Statements like “revolutionize”, “for the first time”, etc. should be avoided.
- The quality of the figures and their readability could be greatly improved.

Further minor comments are provided below.

The title appears overly long for Nat Comm standards.

L. 26 – The meaning of “toughness coordination” is unclear

L. 34 – The manuscript should address the scalability of the proposed approach. Also, it should elaborate on the environmental impacts of this approach and its compatibility with reinforcing elements. Will cement-based materials manufactured with this approach be more or less environmentally friendly than classical cement-based materials, for instance by simply considering the Ref. paste of water-to-cement ratio of 0.5 considered in this work?

L. 84-87 – What is missing here is a sentence about what’s new in this study.

L. 116 – In this study, DI water was used to prepare the samples. What would be the influence of a different chemistry of the water used to prepare cement-based materials? This aspect appears crucial, and an elaboration is considered needed. Essentially, how applicable is this approach to waters that typically contain salts and metals?

L. 122 – Evidence?

L. 123 – Which conditions were considered for this?

L. 126 – Were multiple images like this taken? Did these confirm such results?

L. 131-132 – Where can we see this?

Figure 1 – Also in the figures, such as this one, the English should be corrected

Figure 2 – The sizes of the axis labels are too small to facilitate readability.

Figure 3 a and b – the flexural strength of LLST is higher than the Ref but the specific strength is lower. Why? Also, why is the flexural strength of LLST-2 lower than that of LLST-4?

Figure 3d – It would be useful to specify which type of “strain”. Also, what is the reason for the oscillations in the curves? How many times were these tests repeated? Standard deviation should be provided to prove repeatability.

Figure 3e – A scale is missing here.

Figure 3f – A color legend is missing here.

L. 197 – Evidence? How can you be sure of this?

Figure 3 and 4 – Axis labels are again too small and the quality of the figures could be improved.

Figure 4 h and l – unclear meaning of labels

L. 285-289 – Evidence is needed here. Please expand and justify.

L. 304 – This may be true here, but what about when dealing with larger volumes of materials? This is a critical aspect that should be discussed and supporting evidence should be provided.

L. 315-318 – This may be the case, but this approach is inherently restrained by a problem of scale. Please remark this and elaborate.

- L. 319 – “revolutionized” appears inappropriate here, as similar to other terms used throughout the study.
- L. 365 – Here again, I think it’s essential to expand on the influence of the solution chemistry used to prepare the cement-based material on the gelation results.
- L. 386 – How can this sample size correspond to a representative elementary volume? Please expand.
- L. 387 – How can water be removed at 40°C?
- L. 391 – Was any capping applied?
- L. 398 – How were the samples prepared for the MIP tests? Which drying technique was used? Did the authors check that the drying technique had a negligible influence on the changes in the microstructure of the materials?
- L. 409 – What is the confidence on the fitting performed on the XRD curves? Residuals should be provided to ensure representativeness of the results.
- L. 423 – Clearly, many more ionic species are present at the start of the preparation, though. How does this influence the simulation results? How big of a simplification is this modeling approach?
- L. 439 – Justification for this distance?
- L. 440 – Justification for this box size?
- L. 470 – What’s the rationale/justification for this choice?

Reviewer #3

(Remarks to the Author)

Report on the paper entitled: Nature-inspired low carbon building materials with lightweight, strong and tough properties by tailoring hierarchically porous structure

By Jinyang Jiang, Han Wang, Junlin Lin, Fengjuan Wang, Zhiyong Liua, Ligu Wang, Yali Li, Zongjin Li, Yunjian Li and Zeyu Lu

Submitted for publication in Nature Communications

In this paper, inspired from multiscale hierarchical porous structures found in Nature such as coral or , the authors proposed a low-carbon, lightweight, strong and tough cement-based material (LLST) using an in situ self-assembly strategy, based on the rapid gelation of initial hydrogel. The authors claim an immense improvement of mechanical properties from compressive strength to fracture toughness accompanied by a significant reduction of the global matrix density. Compared with the normal cement paste, LLST displayed a 54% reduction in density, 145% and 1460% improvement in specific compressive strength and fracture energy, accompanied by reduced carbon emission.

Recommendation: For all the reasons listed here below and especially the overstating claim of in-situ self-assembly, I think this paper is not suitable for publication in Nature Communications in its present form.

1/ Self-assembly in Material Science: Indeed, in situ self-assembly has emerged as a promising way to fabricate porous materials with fine-grained pore structure at both micron- and nano-scales, that can be achieved by the spontaneous assembly of small particles driven by chemical reactions. But this is possible in very specific cases such as shown in:

- ref 23 in the context of lithium battery durability where a facile fabrication of in situ self-assembled organic/inorganic hybrid compounds with ordered dual-layer structure was proposed to suppress/disordered/fractal dendritic growth: ordered organic moieties could in situ self-assemble on the metallic lithium surface hence forming an ordered dual-layered interphase structure. This is very far from cement chemistry that proceeds through a dissolution of clinker grains in high pH conditions followed by the precipitation of the hydrate phase called CSH after a solubility threshold has been reached.

- In ref 24 of the context of porous polymers with well-defined multiscale pore morphologies using a solvent-induced self-assembly strategy to synthesize hierarchical porous polymers with tunable morphology, mesopore and nanopore structures using the Friedel–Crafts reaction between polymeric compounds and varying solvent polarity to tailor nanoparticles aggregation. Again, this is completely different than cement chemistry.

- In ref 25 in the context of porous materials being obtained from templating copolymeric phases with the idea to eventually obtaining bicontinuous-structured porous materials. using block copolymers self-assembly with pore sizes and lattice parameters ranging from 1 to 500 nm. Indeed this has opened the route to the synthesis of new families of ordered, pore-size-controlled-by-design porous materials such as MCMs, SBAs, etc... Again, this is completely different than cement chemistry as this route will be inoperative in the context of cement paste setting where pH is only controlled by cement anhydrous clinker grains initial chemistry.

Therefore, although the general idea of guiding cement paste toward a more controlled texture with pre-ascribed property is certainly appealing, the inherent complexity of CSH formation casts of lots of doubt on the idea of in situ controlling the self-assembly of CSH grains in a two-step process that would consist in (i) constructing a nanoporous hydrogel skeleton (in 10

min??) using rapid gelation and (ii), cement hydration and setting from the surface of the hydrogel skeleton, gradually filled-up by hydrogel porosity (filler effect). To me this paper rather reports a simple quite obvious filling up of the micron size pores of an organic hydrogel (with surface being hydrophilic with a large OH density per nm²) with CSH grains adsorbing onto its surface due to obvious oxide particle (CSH) / OH (hydrogel) electrostatic interactions. Actually, CSH formation was reported as mostly occurring from the clinker grains surface in general cement chemistry literature through a simple oxide-on-oxide favorable interaction. Here, this is probably the same situation where hydrogel surface chemistry is compatible with the growth of CSH.

It seems to me that the claimed self-assembly is actually largely overstated. The paper basically shows CSH grains formation starting from the surface of a hydrogel phase through an obvious adsorption process that is simply due to the surface chemistry of the initial host matrix. Nothing really new as this was already described in CSH setting with CSH grains adsorbing onto clinker surface as it gets through dissolution. Can the authors justify the claimed their self-assembly strategy with cement hydration?

2/ Hydrogel: First, the reasons for using hydrogel is not clear at all. It really requires digging into the paper to realize that this is in fact an organic phase described as Poly(NIPAm-SA). Second, its gelation process is also unclear. The reader gets to know that it involves a solution-to-gel transformation in presence of cement clinker grains. But the straight definition of the sol-gel transition is the transition from the sol state (suspension in a liquid) to the gel state (a form of solid). It concerns materials made up of macromolecular entities capable of bonding from one to the next. In the sol state, the entities can move individually, whereas in the gel state their bonds form a three-dimensional network throughout the material. This transition can be triggered by temperature a decrease or a pH change. It is generally reversible.

It is therefore legitimate to ask whether this observed gelation only results from the setting of CSH grain onto the organic phase surface pretty much as in the carbon-cement compounds (see Pellenq et al, US patent, <https://patents.google.com/patent/US20190218144A1/en>) where carbon grains are also together by CSH set phase. As such, in LLSTs, there will be not real sol-gel transition. The overall process being controlled by cement setting process that is accompanied by a large pH increase (12.7) that produces at the same time a CSH (cement hydrate) phase percolating through out the initial hydrogel network and rigidifying it. This is obviously irreversible hence not obeying to the standard sol-gel transition definition. Can the authors elaborate more on LLSTs fabrication.

3/ Water content: Specifically, a large amount of water seems to be absorbed by hydrophilic groups in hydrogel skeleton via hydrogen force, which promoted the in-situ dissolution of cement clinker particles and the precipitation of CSH grains on the hydrogel skeleton.

It is quite surprising to see in Fig. 5a that after 28 days, there is still clinker C₂S and C₃S phases present. And no trace of CSH in diffractograms. This is very unexpected as the water to cement ratio as given in Fig.1 show an excess of water: it is usually around 50% for neat OPC paste while here it varies from 200 to 50% indicating an excess of water in most samples beside that in sample LLST-4 (w/c=50%). This seems to indicate that there is a problem with the water content management? As water is needed to dissolve anhydrous clinker grains and achieved CSH precipitation (that occurs at around 20 mmol Ca, concentration), what is the rationale behind the water/polymer, water/binder ratios as cement grains dissolution and CSH precipitation need the right amount of water given the hydrogel surface affinity to water?

4/ CSH nucleation theory versus adsorbed water: It is claimed by the authors that, based on the heterogeneous nucleation theory, the nucleation barrier during the formation process of cement hydrate CSH grains can be reduced by the nucleation sites on hydrogel skeleton surface. But at the same time, it is also claimed that adsorbed water on the same surface facilitates CSH formation. One of the key statements of this paper is that both of the water and nucleation sites provided by hydrogel skeleton facilitated the in-situ cement hydration on the hydrogel skeleton.

It is known that CSH grains actually form on clinker grains from the local variation of pH and ion concentrations due to differences in clinker surfaces energy. Ineed, clinker grains C₂S, C₃S, are crystals, hence have different surface energies with respect to the decalcification process that puts Ca ions in solution, depending on surface crystallographic orientations. Clinker grain surface defects (steps, lacunes...) can also be the loci of CSH precipitation. But nucleation of CSH grains onto the hydrogel surface relies an obvious catalyst argument: the surface increasing the encounter probability of reactants. As CSH formation is a ions-in-solutions process, it seems that absorbed surface water provides a way to transport, concentrate and precipitate these species onto the hydrogel surface. Why is there a grain precipitation process rather than the formation of a continuous CSH layer all over the hydrogel pore network?

5/ Mechanical properties: The interesting point of the paper is the improved mechanical properties of LLSTs. But comparison should be made with neat OPC and not with foam cement pastes (Fig 3). The 54% reduction of density (compared to OPC) obviously impacts LLSTs mechanical properties. It has to be weaker than a straight OPC. Furthermore, density reduction as observed on LLSTs means more porosity. LLST displayed a 54% reduction in density: this means that LLST has a density close to that of liquid water. This should be clearly stated and the kind of porosity range that has changed compared to normal cement paste should be indicated.

The real interesting question though is LLSTs resistance to shear, plastic deformations and ultimately to fracture (toughness) compared to neat OPC and foam cement pastes. The authors failed to address these crucial points. Can the authors provide mechanical information beyond compressive strength such as fracture toughness?

6/ Life cycling Assessment: This paper lacks of a proper Life cycling Assessment (LCA) of all involved chemical including the hydrogel phase, the gelation agent, etc... The claim of 64% reduction in CO₂ emission is totally unjustified and probably

wrong facing a full LCA assessment giving the used chemicals involving amine that have known and documented health issues.

The disclosed list involves N-isopropylacrylamide (NIPAm), sodium acrylate (SA), N,N-methylenebisacrylamide (Bis), potassium persulfate (APS), and N,N,N',N'-tetramethylethylenediamine (TEMED). Amines are NOT environmentally friendly. Can the authors evaluate the overall environment impact of LLSTs?

7/ Ab initio metadynamics simulations: The computational part is really naïve as it does not go beyond a very molecular description of involves moieties using ab initio technique.

The claim “there is currently no suitable force field to simulate the solution system containing hydrogel and inorganic aqueous species” is wrong. Authors are advised to get exposed to reactive potential such as ReaxFF that can handle all suitable elements. There are realistic atomic models of CSH readily available and probably model for the hydrogel surface that can be handle in the Lammmps simulation package that implements suitable reactive potentials and different metadynamics techniques.

Ab initio Metadynamics simulation were performed. But this requires defining collective variables to obtain free energy profile that is sampling a small subset of the phase space with no guaranty of sampling the right path. The authors have chosen only the most obvious one the simple two-body distance. They could have explored/build other collective variables involving more than just interatomic pair distance to describe the reactivity between a single Ca and OH hydrogel groups in presence of water (the density of water is not even specified). Figures 6h and 6i need proper explanations. The smallness of the simulation system makes the reader wondering if this is representative of the experimental situation. The art of metadynamics requires a proper sampling of both sides of energy barrier that hence needs to be large enough. Did the authors use well-tempered metadynamics? What energy bias did they use?

“strong interactions including van der Waals... : DFT ab initio calculations are notoriously known for their incapacity in predicting van der Waals interactions. In most ab initio simulation packages, dispersion interactions are included under the form of an empirical correction. Hence, any force field calculation would be more efficient as less computationally expensive allowing better sampling of a set of combination of collective variables. Why not going through this route using for instance ReaxFF force field that has been used in many instances in the realm of cement chemistry?

Version 1:

Reviewer comments:

Reviewer #2

(Remarks to the Author)

The authors' revision work satisfactorily addressed all my comments. Based on this consideration, I recommend this manuscript to be accepted for publication.

Reviewer #3

(Remarks to the Author)

The authors have adequately answered my comments. However, I still have the following questions on the accuracy of the metadynamics simulations in regard to the provided data.

1) Although it is true that the well-tempered metadynamics ensures convergence of the free energy by reducing the height of the gaussians, this is a necessary condition, but it is not sufficient to guarantee that the calculations are converged. In fact, one need to monitor the number of barrier re-crossings events to show that both sides of the barrier are adequately sampled. A bias-factor of 20 pretty large (see table), hence very long simulation are required to converge the calculations. Can the authors give a convincing argument re the quality/convergence of their simulations?

2) Moreover, it is very important that the reactant states don't change throughout the simulation to ensure that after each barrier crossing the same states are sampled. It is also important that the recrossing happens along the same pathway so that the corresponding barrier is explored. (see ref 69 in the present manuscript). Given the fact that atoms may significantly rearrange after the Ca has found a Ow, the free energies calculated here probably have very large errors, that cannot be quantified. It would be revealing to repeat the same metadynamics calculation for one Ca site a number of times using different starting condition, to demonstrate the variability of each data point. Can the authors comment this point in regard to their system?

3) The Gaussians' deposition rate is very fast 30fs, the system would have no chance of relaxing to the new equilibrium state. Although this a common practice for ab-initio MD simulations, it makes the results quite unreliable. A deposition rate in the range 1-2ps would be more appropriate at least for classical MD simulation part. With the chosen dual collective variable, convergence might be hard to get over 200 ps trajectories.

4) I urge the authors to have a look to the work by Silvestri et al. (J Chem Theory Comput. 2022 Oct 11;18(10):5901-5919. doi: 10.1021/acs.jctc.2c00787). Silvestri et al. did not propose a method to increase the efficiency of the calculations, but they suggested a protocol to ensure that the calculated free energy was accurate and, more importantly, referenced to standard conditions, as the experiment should be. Silvestri et al. propose to use a linear cylindrical restraint to ensure that the volume of the bound and unbound states are well-defined, which means it is possible to know what the reference state of

the calculation is. The free energies contributions to add and remove the restraints also need to be computed. Can the authors position their work with respect to Silvestri's scheme?

Version 2:

Reviewer comments:

Reviewer #3

(Remarks to the Author)

I went to the reply letter regarding the computational aspects of the paper. the authors thoroughly replied to all my comments showing a profound understanding of the modeling approach (metadynamics technique) they used. In sum, I now support the publication of this work in Nature Communications.

**Dear Editor and Reviewers:**

The authors would like to thank the Editor and Reviewers for their review of our
work and the constructive comments, which are very helpful in improving the quality
of the manuscript. We have studied every comment carefully and incorporated them
as suggested. Revised portions are highlighted in the revised manuscript with remarks
and responses to the reviewers' comments are shown as below:

**Responds to the reviewers' comments**

**Reviewer 1:**

**Q1:** You are describing an in-situ-polymerization of polyacrylamide with
crosslinkers and acrylic acid, forming to a polymergel, which works as a template for
cement hydration. Such copolymerizations have been observed in the past, the most
important works were done by Cölfen.

**R1:** Thanks for your comments. We have carefully reviewed the works done by
Cölfen et al.¹, who developed a novel C-S-H mesocrystal by highly aligning C-S-H
nanoplatelets interspersed with a polymeric binder. However, there are still several
differences between our current work and Cölfen's work, in terms of research object,
role of polymer and research goals, which are summarized as below:

Table R1. Comparison of the current work with Cölfen's work

	Current work	Cölfen's work
Research object	Hydrogel + cement	Polymer chains + C-S-H gels
Role of polymer	As skeleton to regulate hierarchical porous structure of cement paste	As binders to consolidate C-S-H nanoplatelets
Research goals	Reducing carbon emissions of cement industry by developing lightweight cement-based materials with strong and tough properties	Developing C-S-H/polymer composites with high strength

**Q2:** You describe a higher tensile strength, looking at the pictures and data, I
would expect that this is some kind of fiber-reinforced structure, which provides
similar performance. You also show performance on compressive strength, comparing

with foam cement. Air is important for the strength, air size as well. Unfortunately,
 there is no information about the reference foam cement, where you state a higher
 strength value.

**R2:** Thanks for your kind comment. Firstly, the developed hydrogel-cement
 composites exhibited similar tensile performance of fiber-reinforced concrete, and it
 was the first time to report such a great enhancement in toughness of cement-based
 materials without fiber incorporation. Secondly, according to your comments, the
 density, w/c ratio, composition of foam cement reported in literature were
 supplemented in S-Table 2 as follows:

Table 1. Summary of foam cement properties.

Composition	w/c	Compressive strength (MPa)	Dry Density (kg/m ³)	Foaming materials type	References	
Portland cement, Sand	0.4–0.5	5.5–24.3	1300–1600	Protein-based foam agent	Role of non-reactive powder in strength enhancement of foamed concrete	1
Portland cement, glass powder/thermoplastic powder	0.45–0.75	1.5–10.3	800–1580	Protein-based foam agent	Utilization of recycled waste as filler in foam concrete	2
Portland cement, Sand	0.9–1	1.0–7.0	800–1350	Organic foaming agent	Fresh State Characteristics of Foam Concrete	3
Portland cement, Sand	0.9–1	2.0–11.0	650–1200	Organic foaming agent	Shrinkage Behavior of Foam Concrete	4
Portland cement, Sand, Rubber	0.38–0.5	6.4–18.3	1500–1660	Organic foaming agent	Influence of rubber particles on the properties of foam concrete	5
Portland cement	0.4–0.60	0.1–6	400–800	Protein-based foam agent	Effect of Water-Cement Ratio on Pore Structure and Strength of Foam Concrete	6
Portland cement, Flyash	0.33–0.36	8.2–10.4	970–1307	Organic foaming agent	Material Design and Performance Evaluation of Foam Concrete for Digital Fabrication	7
Portland cement, Slag	0.52	0.4–0.8	150–300	H ₂ O ₂	Preparation and characterization of super low density foamed concrete from Portland cement and admixtures	8
Portland cement, Sand	0.4–0.6	2.0–11	650–1200	Organic foaming agent	A classification of studies on properties of foam concrete	9
Portland cement, Slag, Sand	0.55–0.91	1.1–2.0	975–1132	Protein-based foam agent	Physical and mechanical properties of foam concretes containing granulated blast furnace slag as fine aggregate	10

**Q3:** You describe a nucleating effect of acrylic acids. This is akin to the findings
 of Flatt, who showed that highly charged PCEs show a nucleating effect on Ettringite
 in the pore solution.

**R3:** Thanks for your comments. We have reviewed the works done by Flatt ^{2, 3, 4},
 and found that the mechanism behind on nucleating effect of acrylic acids (current
 work) and PCE (Flatt's work) on cement hydrates were totally different, which was
 summarized as follows:

**Flatt's work:** the nucleating effect of PCEs on Ettringite was attributed to the
 electrostatic adsorption of -COOH and -SO₃H in PCE with the Ca²⁺ and Al³⁺
 dissolved in cement pore solution. In that case, the concentrated Ca²⁺ and Al³⁺ around

PCE promoted the crystallization of Ettringite^{2,5,6}.

**Current work:** the nucleation effect of hydrogel on cement hydrates was
achieved by providing water and heterogeneous nucleation sites during the cement
hydration process. In addition, compared with the -COOH and -SO₃H in PCE, the
main chemical groups in hydrogel were -CO-NH₂ and -COONa, which had lower
electronegativity and electrostatic adsorption effects for Ca²⁺ and Al³⁺.

**Q4:** The effect of water storage is well known for superabsorber, which are also
composed of Polyacrylamides.

**R4:** Thanks for your comments. We totally agree that water storage is the
intrinsic property of polyacrylamides-based hydrogels, which have been widely used
as water internal curing agents to mitigate the autogenous shrinkage of cement-based
materials. However, in our research, polyacrylamides-based hydrogel was used as
template to develop lightweight cement-based materials with hierarchical structure,
instead of aiming to mitigate the autogenous shrinkage. Besides that, in previous
research, the addition of hydrogels/superabsorbent polymers into cement-based
materials has been always accompanied by a decline in mechanical strength of the
composite; however, the strength and toughness of hydrogel-cement composited
developed in this study are greatly improved by well-regulating the pore structure
within nano- and micro-scale, which is totally different with the previous research.

**Q5:** The polymerization is not properly described and evaluated. **(1)** APS does
not start polymerization at room temperature. You have to heat, usually 70-100°C.
You don't describe that. This impacts hydration of cement. **(2)** Not polymer analytics
have been done despite some pictures.

**R5:** Thanks. The above concerns are addressed one by one as follows:

(1) Polymerization temperature: APS used in the current research was supplied
by Shanghai Macklin Biochemical Co., Ltd. (99.99% metals basis). In our previous
research, as well as other studies, this type of APS can react at room temperature or
ice bath conditions (seen in Fig. R1)^{7, 8, 9}. The LLST was fabricated at room

temperature, which was noted in the revised manuscript based on your suggestions.

Preparation of PAM/CNS NC gels. The PAM/CNS NC gels were prepared by *in situ* free-radical polymerization. The monomer AM, the initiator APS and the accelerator TEMED were added into the CNS suspension of composition: CNS suspension/AM/APS/TEMED = 60 g/15 g/0.03 g/48 μ l. For the complete reaction of CNS and APS, the mixture was kept at 0 °C in an ice bath for at least 72 h. The polymerization process proceeded in a vacuum environment (0.01 atm) at 0 °C. (G.X. Sun, 2016, Nature Communication)

Fig. R1. Report on the reaction of APS under ice bath conditions.

(2) Polymer analytics: According to your suggestion, the molecular formula of

raw materials and the schematic diagram of the polymerization process were added to

supplementary file.

[REDACTED]

Figure 6. The fabrication process of LLST.

In addition, according to your suggestions, the influence of cement pore solution

on the gelation of hydrogel was investigated. The simulated cement pore solution was

prepared according to “*Recommendation of RILEM TC 260-RSC*”. The results in S. 10.

indicated that the gelation of hydrogel can be accomplished in cement pore solution.

However, compared with DI water, the gelation rate of hydrogel was delayed. This

was because the $S_2O_8^{2-}$ from APS (potassium persulfate) was consumed by OH^- in

cement pore solution⁹. Additionally, water absorption test revealed that the water

absorption rate of hydrogel was reduced by 27% compared to DI water. This

reduction was attributed to interactions between Ca^{2+} and Al^{3+} in cement pore solution

and the anionic groups (-COOH, -OH) in hydrogel, leading to the structure collapse

and decreased content of hydrophilic groups of hydrogel.

S. 10. Gelation of hydrogel in cement pore solution and deionized water.

Q6: Calorimetry has not been described. (1) I cannot see the reaction heat of the polymer. (2) The hydration of LLST1 looks a bit high, having the least cement. (3) For concluding on strength, you would need cumulative heat. (4) Why does the reference cement has no sulfate depletion peak?

R6: Thanks for your suggestions.

(1): Reaction heat of the polymer is represented by the yellow line in Fig. 4 (e); however, compared with heat release of cement hydration, that of polymerization is too small to be seen in Fig. 4 (e). Based on your suggestions, this curve was highlighted in Fig. 4 (e) and noted as follows:

Line 229: The heat flow of LLST, Ref. and hydrogel within 7 days were shown in Fig. 4 (e).

Figure 4 (e). Heat flow of LLST, Ref. and hydrogel within 7 days.

(2): In this research, the Y axis represented the heat release per gram of cement instead of the heat release of total cement. In that case, although LLST1 had the least cement content, the highest w/c ratio of LLST1 contributed to the largest heat release per gram of cement due to more water can be used for reacting with cement particles.

(3): The cumulative heat result was added in supplementary file as follows:

S. 9. (a) types of cement hydrates after 7 days of cement hydration, (b) formation of C-S-H
on hydrogel skeleton, (c) total heat within 7 days of cement hydration.

(4): The sulfate depletion peak was not obvious in the dotted line format. Based
on your suggestion, this curve was redrawn with solid line, as shown in Fig. 4 (e).

**Figure 4 (e).** Heat flow of LLST, Ref. and hydrogel within 7 days.

**Q7:** Degree of hydration: On which basis has this been calculated? Average
phase composition? C₃S only?

**R7:** The cement hydration degree was calculated by the content variation of
clinker phase (C₃S., C₂S, C₄AF and C₃A) according to QXRD results. For this test,
the internal standard corundum was adopted with full Rietveld analysis via Topas3-C
software. Based on the reviewer's suggestions, we added extra explanation in the
methods part as follows:

Line 441: Cement hydration degree was calculated based on the content of
clinker phase (C₃S, C₂S, C₄AF and C₃A) obtained in QXRD.

**Q8:** Particle size of ettringite has a huge effect on tensile strength of cement. You

have to consider in the discussion.

**R8:** Thanks for your suggestions, which provide further insight into the strength
development of LLST. We have added this to the discussion as follows:

**Line 209:** SEM images revealed needle-like and hexagonal plate-like cement
hydrates with large crystal sizes in LLST (Fig. 4b&c), resulting from that the
sponge-like porous structure offered sufficient space for the growth of these cement
hydrates. Notably, the **needle-like cement hydrates (AFt/AFm) in larger size were**
**expected to play similar roles as fiber in preventing the propagation of cracks and**
**contributing to enhanced flexural strength in cement paste.**

**Q9:** What is the chemistry of your cement?

**R9:** Thanks for your kind suggestions. The chemical composition and phase
composition of cement were added in S-Table 8-9 in supplementary file as follows:

S-Table 7. Phase composition of cement.

Cement	C ₃ S	C ₂ S	C ₄ AF	C ₃ A
	62.47	16.85	5.51	12.58

S-Table 8. Chemical composition of cement.

	CaO	SiO ₂	Al ₂ O ₃	SO ₃	Fe ₂ O ₃	LOI
Cement	63.88	21.31	5.67	2.32	3.74	1.53

**Q10:** I don't understand how you calculate CO₂. Please provide the underlying
data and considerations you used.

**R10:** Thanks for your kind suggestions. According to “*ISO 14040:*
*Environmental management-Life cycle assessment-Principles and framework*”, the
calculation details of CO₂ emission were added in the supplementary file, including
the calculation formula, parameter and data resource.

S-Table 2. Different cases of LCA calculation.

Mix. (ton)	LLST-1	LLST-2	LLST-3	LLST-4	Reference (Ref)
Cement	0.43	0.75	1	1.2	1.2
NIPAm	0.048	0.048	0.048	0.048	-
SA	0.012	0.012	0.012	0.012	-
Bis	0.0009	0.0009	0.0009	0.0009	-
APS	0.0009	0.0009	0.0009	0.0009	-
TEMED	0.00072	0.00072	0.00072	0.00072	-

6.3. Life cycle inventory

The global warming potential (GWP), acidification potential (AP), eutrophication potential (EP) of raw materials was considered. It should be noted that the emissions of cement includes raw materials decomposition, calcination fuel, transportation fuel and electric power. The emissions of organic includes two parts: electric power and transportation fuel.

S-Table 3. Emissions from the production of raw materials.

Mix. (per ton)	GWP (kg CO ₂)	AP (kg SO ₂)	EP (kg NO _x)	Reference
Cement	9.40 E+02	5.00 E-01	1.53 E+00	
NIPAm	3.1 E+03	7.10 E+00	5.08 E-1	13
SA	2.6 E+03	8.49 E+00	1.01 E-00	16,17,18

S-Table 4. Emissions from the transportation of raw materials.

Mix. (per ton)	GWP (kg CO ₂)	AP (kg SO ₂)	EP (kg NO _x)	Reference
Road transportation (Diesel)	1.63 E+02	2.00 E+00	1.09 E+00	19, 20

Note: the transport distance here was set as 100 km.

S-Table 5. Emissions from the production of LLST.

Mix. (per m ³)	GWP (kg CO ₂)	AP (kg SO ₂)	EP (kg NO _x)	Reference
LLST products	3.05 E+00	-	-	

The details of the LCA of LLST cement are displayed as follows, which is mainly includes four formulas:

$$C_{je} = C_{xc} + C_{ys} \dots \dots \dots (1)$$

C_{je} : Emissions during the production and transportation of each building materials
 C_{xc} : Emissions from production stage of each building materials (kgCO₂e);
 C_{ys} : Emissions from transportation stage of each building materials (kgCO₂e);

$$C_{xc} = \sum_{i=1}^n M_i \cdot F_i \dots \dots \dots (2)$$

M_i : Consumption of each building materials (ton);
 F_i : Emission factor of LLST per unit weight (kg/ton). Emissions from the production of LLST (E_{llst}) were also taken into F_i , according to the formula (3):

$$F_{ip} = \frac{E_{ip} \cdot k_{ip}}{N} \dots \dots \dots (3)$$

E_{ip} : Emissions from the production of LLST
 E_{ip} : The total amount of electricity consumed
 k_{ip} : Emission factor of electric energy (around 0.86 (kgkW_h))
 N : Total amount of LLST productions

$$C_{ys} = \sum_{i=1}^n M_i \cdot D_i \cdot T_i \dots \dots \dots (4)$$

M_i : Consumption of each raw materials (ton);
 D_i : Average transport distance of each raw materials (km);
 T_i : Emission factor of each raw materials per unit weight and per unit transportation distance (kgCO₂e/(ton km)).

6.5. Interpretation of Results

S-Table 6. Emissions from the LLST.

Mix. (ton)	GWP (kg CO ₂)	AP (kg SO ₂)	EP (kg NO _x)
LLST-1	684.79	1.51768	1.16656
LLST-2	1037.75	2.31768	2.00496
LLST-3	1313.5	2.94268	2.65996
LLST-4	1534.1	3.44268	3.18396
Reference (Ref)	1354.1	3	3.144

In summary, we tried our best to improve the manuscript and made some
 changes with remarks in the revised manuscript. We appreciate for the Reviewer's
 kind comments and questions and hope that the reply will meet your approval. Finally,
 we would like to express our great appreciation for your valuable suggestions.

**Reviewer 2:**

**Q1:** The novelty of this work relative to the state-of-the-art is poorly justified.

**R1:** Thanks for your comments. We apologize for failing to elaborate on the
innovations as compared with literature. In fact, two new points, to our knowledge,
were proposed in the current study, as follows:

**(1):** Normally, the pores in cement matrix were uncontrollable, varying randomly
in size and shape. However, a novel method was developed in our research that can
control the shape and size of pores using hydrogel. This approach resulted in a
sponge-like pore structure of LLST with hierarchical sizes (micropores: 1~50 μm ,
nanopores: 5~100 nm). More importantly, this approach showed the potential for
active design of concrete structures by using hydrogel with different structures,
therefore enhancing the long-term serviceability of concrete materials in diverse
application scenarios (high load, high strain or high vibration).

**(2):** In addition, the water in cement was transformed from liquid into solid state,
which cannot be achieved before. This method controlled the distribution pattern of
water and its reaction time with cement through the use of hydrogel. It offered new
insights into microstructure construction and hydration rate control, which contributed
to the programming design of cement structures and properties. Furthermore, this
method overcame the limitation bleeding effects on water content, which introduced
more water into cement matrix. In that case, the cement hydration degree can be
increased. At the same time, after the water evaporated, more pores can be formed in
the matrix.

The novelty of this work had been emphasized in the revised manuscript as
follows:

**Line 121:** Normally, the pores in cement matrix are uncontrollable, varying
randomly in size and shape. However, as shown in Fig. 1d and S. 1 (c-f), it was
observed that all the LLST showed sponge-like and hierarchical porous structures
through a rapid gelation of hydrogel as skeleton and subsequent deposition of cement
hydrates as skin in order.

**Line 144:** Compared with *Ref.*, a significantly higher volume of 5-100 nm

nanopores was present in all LLSTs. This was because the water absorption behavior
of the hydrogel allowed more water to mix evenly with the cement particles,
preventing the bleeding of water in the slurry due to density differences.

**Q2:** The study makes several statements that are poorly justified. This issue
should be fixed, either by providing results or adequate references in support of such
statements. Without this groundwork, parts of this study appear inadequately
supported by data.

**R2:** Thanks for the suggestion. We made great effects to ensure the statements
were well-supported by data or references. Those unsupported claims were revised or
removed in line 80, line 121, Line 144 and Line 212.

**Q3:** The study is inherently limited by a problem of scale, as extremely small
samples are prepared and tested. What is the applicability of this approach to larger
volumes of materials. Without proven applicability, this work may remain only a
theoretical exercise, as no structures made of LLST may be built in practice in the
future.

**R3:** Thanks for your suggestions. We also prepared a series of LLST samples
with the standard testing size of 40×40×40 mm and 40×40×100 mm for the
compressive and flexural strength measurement, based on GB/T 17671-2021. The
results were consistent with the small samples in the current study. We believe this
novel strategy can be used to apply in a large-scale application once the “magnetic
stirring condition” can be satisfied at project site. More importantly, the fabrication
process of the LLST is really convenient and time-saving, requiring only 10 minutes
of stirring after mixing, which is more practical compared to the method involving
directional freezing ¹⁰ (Yang Zhou, Nature Communications, 2023), high pressure ¹¹,
high temperature ¹², or specialized equipment ¹³.

**Q4:** The experiments use DI water to prepare the considered samples. What is
the influence of ionic species that are always present in the water used to prepare

cement-based materials on the results of this work?

**R4:** Thanks for your kind comments. We believe the influence of ions in tap
water on LLST performance can be ignored. Compared with DI water, the tap water
normally used to prepare cement-based materials contains extra ions, including Ca^{2+} ,
Na^+ , Mg^{2+} , K^+ , CO_3^{2-} and SO_4^{2-} (5 ~ 80 mg/L). However, these ions would also be
released by cement particles (>1000 mg/L) dissolved in DI water ¹⁴, and the
corresponding ions concentration would be much higher than that in tap water. In that
case, the influence of ions in tap water on LLST performance can be ignored. In our
ongoing research, the strength of LLST prepared with tap water indeed showed no
difference from that prepared with DI water.

**Q5:** Several testing features and modeling assumptions must be provided, as
they have been presently omitted.

**R5:** Thanks for the suggestion. We appreciate the opportunity to clarify this part.
More details about testing and modeling assumptions were added in the revised
manuscript in line 452-676.

**Line 394:** Before the test, the mixture was placed in a curing container with a
temperature of 20°C and RH of 95%.

**Line 402:** 7-day and 28-day specimens (20 mm cube for compressive test and
20×20×80 mm cuboid for flexural test) were dried in a forced ventilation oven at
40°C for 24 hours to remove water. Before the test, the load surfaces of specimens for
compressive strength test were capped.

**Line 408:** The fracture toughness of specimens were detected followed by the
draft recommendation proposed by RILEM Committee based on the fictitious model.
The specimens (20×20×80 mm) were cut at mid-span with a half depth. The loading
rate for thistest was set as 0.02 mm/min.

**Line 425:** After 28 days of curing, the samples for MIP test were frozen in liquid
nitrogen for 10 minutes, and then dried in a lyophilizer below -50oC for 24 hours.
After that, LLST was split into small pieces around 2mm. 1H Nuclear Magnetic
Resonance Spectroscopy (NMR) (MesoMR12-060V-I, Niumag, China) was also used

in this research to detect the porosity of of specimens. Before the NMR test, LLST
specimens (20 mm cube) was put in a vacuum saturated device for 24h to achieve
water-saturated state,

Line 441: Cement hydration degree was calculated based on the content of
clinker phase (C_3S , C_2S , C_4AF and C_3A) obtained in QXRD¹⁵. In addition, TAM Air
isothermal calorimeter (TA Instruments, USA) was used to measure the heat of
hydration for 7 days at 25°C. To maintain a consistent temperature, all raw materials
were stored in a thermostatic chamber set to 20°C for 24 hours before the test. The
heat evolution of LLST, Ref. and hydrogel within 7 days were recorded.

Line 451: The porous solid phase of hydrogel in LLH cement is a copolymer
system, which is composed of NIPAM and SA monomers as well as Bis molecules (S.
6). During the process of in situ polymerization of monomers with cement hydration,
a copolymer skeleton is firstly constructed, followed by the coordination of the metal
ions (mainly Ca^{2+}) released by cement particles by carboxyl group from hydrogel⁴⁹.
Thus, it is more likely that the site of interactions between hydrogel and the ions
released by cement particles is in the pore solution environment. Based on this
promise, the model should be constructed by putting polymer in solutions containing
water molecules, Ca^{2+} , and SiO_4^{2-} ions. Since the interaction between the Ca ion and
the carboxyl group is the primary reaction between the hydrogel and aqueous species
in the cement pore solution, we isolate the local structure of the hydrogel composed of
one NIPAM and one SA unit (with the Na ion removed) for this model. The
NIPAM-SA monomer, together with a single Ca ion, is placed within an explicit
solvation shell with a hydration radius of 5.5 Å (the density of water is 1g/cm³). This
simplified model is intended to generate a training dataset for the machine learning
force field (MLFF) using ab initio molecular dynamics (AIMD) simulations. The final
model consists of 481 atoms within an orthogonal simulation box with dimensions of
16.6 Å × 17.2 Å × 17.6 Å.

**Q6:** The English language should be revised in detail by a native speaker, as
many grammatical errors arise from the start of the abstract (e.g., L. 18 is unclear; L.

23 presents an error; etc.). Statements like “revolutionize”, “for the first time”, etc.
should be avoided.

**R6:** Thank you for your valuable feedback. The manuscript has been thoroughly
polished by a native English speaker. The revisions were highlighted in the revised
manuscript.

**Q7:** The quality of the figures and their readability could be greatly improved.

**R7:** Thanks for your kind suggestions. The resolution of Fig. 2 ~ Fig. 6 was
improved, and the important information was highlighted in the revised manuscript.

**Q8:** The title appears overly long for Nat Comm standards.

**R8:** Thanks for pointing that out, the title was replaced as follows:

**Nature-Inspired Hierarchical Building Materials with Low CO₂ Emission and**
**Superior Performance**

**Q9:** L. 26 – The meaning of “toughness coordination” is unclear

**R9:** Thanks for your suggestions. “toughness coordination” was used to indicate
that LLST has both high strength and toughness, which was inappropriate in the
sentence. This phrase was removed in the revised manuscript.

**Q10:** L. 34 – The manuscript should address the scalability of the proposed
approach. Also, it should elaborate on the environmental impacts of this approach and
its compatibility with reinforcing elements. Will cement-based materials
manufactured with this approach be more or less environmentally friendly than
classical cement-based materials, for instance by simply considering the Ref. paste of
water-to-cement ratio of 0.5 considered in this work?

**R10:** Thanks for your suggestions. The **life cycle assessment** (LCA) of LLST
was evaluated to elaborate on the environmental impacts of this approach. The Global
Warming Potential (GWP), Acidification Potential (AP) and Eutrophication Potential

(EP) of LLST were calculated based on the CO₂, SO₂ and NO_x emission during the
raw material production, transportation, and product processing. The results indicated
that GWP, AP and EP of LLST1 were reduced by 49%, 49% and 63%, respectively, as
compared with that of Ref. paste (w/c=0.5), so it is convincing that the developed
LLST is environmentally beneficial.

**Q11:** L. 84-87 – What is missing here is a sentence about what’s new in this
study.

**R11:** Thanks for pointing that out, we added the relevant sentence as follows:

**Line 80:** Nevertheless, the controlled design on the shape and size of micro and
nano pores in cement-based materials has not been previously investigated within the
relevant field.

**Q12:** L. 116 – In this study, DI water was used to prepare the samples. What
would be the influence of a different chemistry of the water used to prepare
cement-based materials? This aspect appears crucial, and an elaboration is considered
needed. Essentially, how applicable is this approach to waters that typically contain
salts and metals?

**R12:** Thanks for your kind comments. This approach is also applicable to waters
that typically contain salts and metals. This is because, cement particles would
immediately release several ions (Ca²⁺, Na⁺, Mg²⁺, K⁺, CO₃²⁻ and SO₄²⁻ (>1000 mg/L)
into fresh cement slurry of LLST after being mixed with water. The ions released by
cement particles have stronger effect on the hydrogel than the salts and metals
contained in water. In that case, the influence of salts and metals in water on LLST
performance can be ignored.

**Q13:** L. 122 – Evidence?

**R13:** According to your suggestion, the explanation about this sentence and
relevant references were added in this part of the revised manuscript, as follows:

**Line 110:** In that case, the cement hydrates tended to form around the network

skeleton. Additionally, based on the heterogeneous nucleation theory^{28, 29}, the
nucleation barrier during the formation process of cement hydrates can be reduced by
the nucleation sites from hydrogel skeleton. This phenomenon also facilitated the in-
situ cement hydration on the hydrogel skeleton.

**Q14:** L. 123 – Which conditions were considered for this?

**R14:** Thanks for your question. The LLST was cured in a container at 20 °C and
95% RH. The curing conditions for LLST were added in the revised manuscript.

**Line 394:** Before the test, the mixture was placed in a curing container with a
temperature of 20°C and RH of 95%.

**Q15:** L. 126 – Were multiple images like this taken? Did these confirm such
results?

**R15:** Thanks for your question. More than 40 images were taken for this
phenomenon, which proved that the micropores in hydrogel skeleton were filled by
uniform dispersed cement hydrates. We added more representative images in S.1 (d-f)
of supplementary file to show this process as suggested.

**S. 1.** Microstructure of pure hydrogel and LLST at different hydration ages, (a) 10 min after
 fabrication (b) after 7 days hydration, (c) after 28 days hydration, (d-f) Process of hydrogel
 skeleton gradually filled by uniform dispersed cement hydrates.

**Q16:** L. 131-132 – Where can we see this?

**R16:** Thanks for your question. Extra SEM results were added in supplementary
 file to support this statement ‘sponge-like hierarchical porous’, as shown in the
 supplementary file (S. 1 (b-c)). The phenomenon of “hierarchical porous structure
 helps to mitigate the stress concentration and realize uniform stress distribution” was
 reported by ¹⁶, and was also added in the revised manuscript, as seen in Line 126.

**Q17:** Figure 1 – Also in the figures, such as this one, the English should be
 corrected.

**R17:** Thanks for pointing that out. The description and grammar in Fig. 1 legend
 were corrected, as follows.

**Figure 1.** Microstructure analysis at different hydration ages by SEM, (a) the microstructure
of LLST after 10 min of fabrication, (b) the microstructure of LLST after 7-day curing, (c-d) the
microstructure of LLST after 28-day curing.

**Q18:** Figure 2 – The sizes of the axis labels are too small to facilitate readability.

**R18:** Thanks for your kind reminder. We redraw the Fig. 2 based on your
suggestion, as seen below:

**Figure 2.** Pore structure of LLSTs, (a) spatial distribution of micropores in LLST-1 by X-CT,

(b-c) size distribution and cumulative volume of nanopores by MIP, (d) schematic hierarchical
micro/nanoporous structure.

**Q19:** Figure 3 a and b – the flexural strength of LLST is higher than the Ref. but
the specific strength is lower. Why?

**R19:** We apologize for the unclear description of Fig. 3b, which was the specific
compressive strength compressive strength of LLST rather than flexural strength. In
Fig. 3b, the specific strength of LLST4 was lower than Ref. This was because the w/c
ratio and porosity of LLST4 and Ref. are similar, the beneficial effects of hydrogel on
strength through introducing more water and pores have not been realized in LLST4.
At the same time, LLST4 had the lowest water content in LLST samples, which was
not conducive to the gelation of hydrogels and thus inhibited their toughening effect.

**Q20:** Also, why is the flexural strength of LLST-2 lower than that of LLST-4?

**R20:** Thanks for your question. We checked the raw data, and found that the
The lower flexural strength of LLST2 was related to an abnormal result of one
sample. As shown in Fig. R1, the crack offset during the testing resulted in a very low
strength value for this sample. We did not notice that, and it caused the reduction in
the average strength of LLST2. We re-prepared and tested the compressive and
flexural strength of LLST2, and re-draw the Fig. 3 (a, b and d).

**Fig. R1.** The formation of crack offset in LLST2 sample.

**Q21:** Figure 3d – It would be useful to specify which type of “strain”. Also, what
is the reason for the oscillations in the curves? How many times were these tests

repeated? Standard deviation should be provided to prove repeatability.

**R21:**

(1): We apologize for the unclear description of Fig. 3d, which should be
compressive strain in X axis. This has been corrected in the revised manuscript.

**Figure 3.** Mechanical properties of LLSTs, (a) 28-day flexural strength, (b) specific
compressive strength and (c) a comparison with reported foam cement, (d) strain-compressive
stress curve of LLSTs, (e-f) schematic diagram of multi-scale stress dispersion.

(2): Thank you for your comment. As observed in the compression curves in Fig.
3d, the stress initially undergoes a monotonic increase until the first peak is reached,
indicating crack initiation from intrinsic defects. Following this, the curves exhibit
clear oscillations, with some showing a slow softening trend (where stress decreases
gradually) and others demonstrating a hardening trend (where stress rises again to a
higher level).

This phenomenon can be explained as follows: in a traditional cement matrix,
during the compression process, the presence of defects alters the stress state in the
vicinity from compression to tension or shear, driving crack propagation. As these
cracks penetrate the cement matrix, they interact with multiple defects, leading to
overall failure characterized by a rapid reduction in stress on the strain-stress curve.

In contrast, in LLST, the hydrogel skeleton effectively bridges the cracks. When
the cracks attempt to open or slide, stretching the hydrogel or breaking the interface

between the cement matrix and the hydrogel is required, necessitating greater force.
 This, in turn, results in the oscillations observed in the strain-stress curve.
 Additionally, compared to conventional cement paste, the hydrogel network refines
 the defect structure by reducing the presence of large defects in the matrix. This helps
 prevent stress concentrations and localized failures associated with those large defects.
 Therefore, the strain-compressive stress curve exhibits oscillations rather than a rapid
 decrease with increasing strain.

(3): This test was repeated three times for each sample. Based on your
 suggestions, additional strain-stress curves of LLST were provided in S.3, as follows:

 **S. 3.** (a) Midspan deformation rate and fracture energy of LLST, (b-c) Load-displacement
 curve of *Ref.*, and LLST, (d) Strain-compressive stress curve of LLSTs.

**Q22:** Figure 3e – A scale is missing here.

**R22:** Thanks for pointing that out, and the scale bar was added in Fig. 3e, as
 follows:

Figure 3 (e). schematic diagram of multi-scale stress dispersion.

Q23: Figure 3f – A color legend is missing here.

R23: Thanks for pointing that out, and the color legend was added in Fig. 3f as follows:

Figure 3 (f). schematic diagram of multi-scale stress dispersion.

Q24: L. 197 – Evidence? How can you be sure of this?

R24: Thanks for your question. This was inappropriate, and we removed it from the manuscript.

Q25: Figure 3 and 4 – Axis labels are again too small and the quality of the figures could be improved.

R25: Thanks for pointing that out, and the axis labels were redrawn in Fig 3 and 4, as follows:

**Figure 3.** Mechanical properties of LLSTs, (a) 28-day flexural strength, (b) specific
 compressive strength and (c) a comparison with reported foam cement, (d) Compressive
 strain-strength curve, (e-f) schematic diagram of multi-scale stress dispersion.

**Q26:** Figure 4 h and I – unclear meaning of labels

**R26:** The label with clear description has been added for clarity in the revised
 manuscript.

**Q27:** L. 285-289 – Evidence is needed here. Please expand and justify.

**R27:** Thanks for your suggestion. We added explanation about this part with
 more references to support, as follows:

Line 215: The growth pattern of cement hydrates in LLST can be explained as
 follows: on the one hand, the water-rich environment near the hydrogel skeleton
 facilitated the dissolution of cement particles and the formation of cement hydrates;
 on the other hand, the hydrogel skeleton played a role as hydration nucleation sites,
 which promoted the crystallization of cement hydrates on its surface.

**Q28:** L. 304 – This may be true here, but what about when dealing with larger
 volumes of materials? This is a critical aspect that should be discussed and supporting
 evidence should be provided.

**R28:** Thanks for your kind comments. In our research, the heat release of LLST

was confirmed by isothermal calorimeter characterization. According to previous
research ¹⁷, **the results obtained by isothermal calorimeter can reflect the heat**
**release of large-volume samples**, consistent with results obtained through
monitoring systems or simulation software (COMSOL Multiphysics)¹⁸. Based on your
suggestions, additional explanations were added to demonstrate the relevance between
small-scale experimental samples and large-volume concrete in practical engineering.

Line 237: Normally, the lower heat release of LLST benefits the internal
temperature reduction of concrete in the early stages of practical engineering.
Therefore, such a unique water supply mode in LLST was believed to positively
control the cement hydration reaction and reduce the risk of early-age thermal
cracking in mass concrete.

**Q29:** L. 315-318 – This may be the case, but this approach is inherently
restrained by a problem of scale. Please remark this and elaborate.

**R29:** Thanks for your suggestion. The limitations of small-sized samples in this
part have been declared in the revised manuscript as follows:

Line 254: Notably, there is a discrepancy between the sample size in this study
and those encountered in practical engineering applications, which should be
addressed in further research.

**Q30:** L. 319 – “revolutionized” appears inappropriate here, as similar to other
terms used throughout the study.

**R30:** Thanks for pointing that out, and this word was removed.

**Q31:** L. 365 – Here again, I think it’s essential to expand on the influence of the
solution chemistry used to prepare the cement-based material on the gelation results.

**R31:** Based on your comment, the influence of cement pore solution on the
gelation of hydrogel was investigated. The simulated cement pore solution was
prepared according to “*Recommendation of RILEM TC 260-RSC*”. The results in S. 10.
indicated that the gelation of hydrogel can be accomplished in cement pore solution.

However, compared with DI water, the gelation rate of hydrogel in alkaline cement
solution was delayed. This was because the $S_2O_8^{2-}$ from APS (potassium persulfate)
was consumed by OH^- in cement pore solution ⁹. Additionally, water absorption test
revealed that the water absorption rate of hydrogel was reduced by 27% compared to
DI water. This reduction was attributed to interactions between Ca^{2+} and Al^{3+} in
cement pore solution and the anionic groups ($-COOH$, $-OH$) in hydrogel, leading to
the structure collapse and decreased content of hydrophilic groups of hydrogels. This
part was added in supplementary files as section 5.

S. 10. Gelation of hydrogel in cement pore solution and deionized water.

**Q32:** L. 386 – How can this sample size correspond to a representative
elementary volume? Please expand.

**R32:** Thanks for your suggestions. Based on your suggestions, the compressive
529 /flexural strength of LLST samples in large size (40×40×40 mm) and (40×40×160
530 mm) were tested. The results were consistent with those obtained on small scales,
which proved the applicability of LLST in practical application.

**Q33:** L. 387 – How can water be removed at 40°C?

**R33:** Thanks for your questions. We apologize for the lack of information on
humidity. Samples drying was conducted in a forced ventilation oven ($RH < 30\%$, 40
536 °C), in which the air convection facilitated the transfer of moisture from the LLST to
537 air. Based on your suggestions. Details about the oven conditions have been added to
538 the revised manuscript.

**Line 402:** 7-day and 28-day specimens (20 mm cube for compressive test and
20×20×80 mm cuboid for flexural test) were dried in a forced ventilation oven at 40
541 °C for 24 hours to remove the inside water.

**Q34:** L. 391 – Was any capping applied?

**R34:** Thanks for your questions. The sample was capped before the test. We
noted it in the revised manuscript as follows:

**Line 404:** Before the test, the load surfaces of specimens for compressive
strength test were capped.

**Q35:** L. 398 – How were the samples prepared for the MIP tests? Which drying
technique was used? Did the authors check that the drying technique had a negligible
influence on the changes in the microstructure of the materials?

**R35:** Thanks for your questions. The samples for MIP test were prepared as
follows:

**Line 425:** After 28 days of curing, the samples for MIP test were frozen in liquid
nitrogen for 10 minutes, and then dried in a lyophilizer below -50°C for 24 hours.
After that, LLST was split into small pieces around 2mm.

According to ¹⁹, the freeze-drying method causes less damage to the pore
structure of cement paste compared with the other common drying methods, i.e. oven
drying and vacuum drying.

**Q36:** L. 409 – What is the confidence on the fitting performed on the XRD
curves? Residuals should be provided to ensure representativeness of the results.

**R36:** Residual weighted profile (Rwp) was used as an indicator of the degree of
software fitting and result accuracy. In this study, the values of Rwp were in the range
of 7~10, which was below the threshold of 15. Therefore, the results of Topas 4.2
calculations were credible. Based on your suggestion, the Rwp was added in the
manuscript files as follows:

**Line 441:** The residual weighted profile value in all specimens was in the range

of 7~10.

**Q37:** L. 423 – Clearly, many more ionic species are present at the start of the
preparation, though. How does this influence the simulation results? How big of a
simplification is this modeling approach?

**R37:** Thank you for your insightful question. You're correct that the cement pore
solution contains a variety of ionic species during the experimental preparation process.
However, in the simulation section, we focus specifically on the molecular mechanism
of the interaction between the Ca ion and the carboxyl group of the hydrogel. The
reason is that the carboxyl group is the only reactive site on the hydrogel that can
interact with ionic species in cement pore solution, and the Ca ion, being the most
abundant species produced by cement dissolution, has the greatest reactivity of
interacting with the carboxyl group.

As such, the interaction between the Ca ion and carboxyl groups is the dominant
reaction between the hydrogel and aqueous species in the cement pore solution. Other
species, such as Na ions and K ions, have a negligible influence on this reaction due to
their lower reactivity with the carboxyl group. As for the silicate species, it is unlikely
to directly interact with the hydrogel, as there are no reactive sites for such interactions
in the hydrogel structure.

To ensure computational efficiency, we simplified the system by focusing
exclusively on the Ca²⁺-carboxyl interaction, excluding other ionic species. While this
simplification reduces the complexity of the system, it does not compromise the
accuracy of our results. The modeling approach we employed is machine learning force
field molecular dynamics (MLMD), which is the state-of-the-art method in the
computational physics and chemistry. MLMD combines the sampling capabilities of
traditional MD with the quantum mechanical accuracy of density functional theory
(DFT) calculations. This ensures that the molecular mechanisms we uncover are
reliable and reflective of the underlying physics. The details of the develop the machine
learning force field (MLFF) and the setup of the MLMD were shown in the manuscript
as follows:

The computational details of the construction of the MLFF were shown in the
manuscript as follows:

[revised manuscript text omitted]

 **S. 4. (a) and (b)** Comparison of the energies and atomic forces calculated by the MLFF and DFT
 for configurations in the AIMD-based WT-MetaD training dataset. **(c) and (d)** Comparison of the
 energies and atomic forces calculated by the MLFF and DFT for configurations in the perturbed
 structure training dataset. **(e) and (f)** Comparison of the energies and atomic forces calculated by
 the MLFF and DFT for configurations in the test dataset.

**Machine learning force field molecular dynamics simulations:** All machine
 learning force field molecular dynamics (MLMD) simulations were performed in the

LAMMPS code³⁴ using the fine-tuned model. The periodic boundary conditions were
 applied in all directions. The timestep of 1.0 fs was adopted. The initial configuration
 was first performed energy minimization with conjugate gradient algorithm and then
 equilibrated at the chosen temperature with NPT ensemble for 100 ps. In production
 run, the NVT ensemble was applied, and the temperature was controlled by the
 Nose-Hoover thermostat with a relaxation time of 100 fs. The duration of the
 production run was 1 ns for each simulation.

To comprehensively elucidate the molecular mechanisms underlying the
 interaction between the Ca ion and the carboxyl group, three collective variables were
 chosen for the WT-MetaD simulations. In the simulation 1 (MLFF 1), we aimed to
 determine the most stable configuration for the binding of the Ca ion to the carboxyl
 group. To achieve this, the coordination numbers (CN) of the Ca ion with the O ion
 from water molecules (O_w) along with the CN of the Ca ion with the O ion from
 carboxyl group (O_c) were adopted to form the two-dimensional CVs. The CN was
 computed using the following equation (Eq. 2):

$$687 \quad CN(Ca - O_{w/c}) = \sum_{j \in O_{w/c}} S_{ij}(r_{ij}) = \sum_{j \in O_{w/c}} \frac{1 - \left(\frac{r_{ij} - d_0}{r_0}\right)^n}{1 - \left(\frac{r_{ij} - d_0}{r_0}\right)^m} \quad (2)$$

where r_{ij} is the distance between atom i and atom j . $s_{ij}(r_{ij})$ is a switching function
 describing the coordination between atom i and j . d_0 is the central value of the
 function. r_0 is the acceptance distance of the switching function. The d_0 and r_0 were set
 as 2.42 Å and 0.4 Å, respectively, which were tested to be appropriate^{35, 36}. n and m
 are 6 and 12, respectively^{37, 38}.

In simulation 2 (MLFF 2), the distance between the Ca ion and one of the O_c ions
 was chosen to explore the reaction mechanism as the Ca ion approaches the carboxyl
 group. In simulation 3 (MLFF 3), the distance between the Ca ion and the center of
 mass (COM) of two O_c ions was selected to verify the rationality of the bonding
 mechanism found by different choice of CVs. The parameters of the MLFF-based
 WT-MetaD are shown in Table 2. Convergence tests for free energy surfaces, time
 evolution of the CVs, time evolution of the height of bias are shown in Fig. S. 5-7.
 The errors in the free energies for MLFF 2 and MLFF 3 were calculated using the
 block-analysis method³⁹. The average error converged at the block size of around 900

in both simulations (Fig. S. 8). Using 900 blocks, the average errors were calculated
 to be 1.22 kJ/mol for MLFF 2 and 1.42 kJ/mol for MLFF 3.

**Table 2.** Parameters of WT-MetaD simulations

Project	CV	Quadratic walls		Initial height	Width	Deposition frequency	Biasfactor
		Lower position	Upper position				
AIMD	Distance	1.8 Å	6.8 Å		0.1 Å		
MFL 1	CN(Ca-O _c)	3.5			0.1		
	CN(Ca-O _w)	0.1			0.05		
MFL 2	Ca-O _c			3.5 kJ/mol		30 fs	20
	Distance	2 Å	6.5 Å		0.1 Å		
MFL 2	Ca-COM						
	Distance	2 Å	5.5 Å		0.07 Å		

**Q38:** L. 439 – Justification for this distance? L. 440 – Justification for this box
 size? L. 470 – What’s the rationale/justification for this choice?

**R38:** Thank you for your comments. The choice of a 5.5 Å hydration shell radius of
 the Ca-hydrogel system is to provide enough water molecules that can prevent the
 self-interaction due to the image effect caused by the periodic boundary condition in the
 MD simulations. The 5.5 Å radius of the hydration shell for Ca-hydrogel contains 300
 water molecules, which can model the solvent effects. A larger shell would
 significantly increase computational costs without contributing much additional
 accuracy for this system.

As for the box size (16.6 Å × 17.2 Å × 17.6 Å), it is depended on the size of the
 solvation shell around the NIPAM-SA monomer and Ca²⁺ ion with the aim of
 accommodate all molecules in the system. Besides, we choose this relatively large
 system with respect to the DFT calculation to maintain sufficient space to avoid
 artificial interactions between periodic images. This box size ensures that the system is
 sufficiently isolated, preventing interactions between the ion and the carboxyl group in
 neighboring periodic images, which would distort the simulation results. Additionally,
 the dimensions were chosen to be large enough to capture the relevant solvation and
 coordination behavior of the Ca²⁺ ion while keeping the simulation computationally
 feasible.

In summary, we tried our best to improve the manuscript and made some
changes with remarks in the revised manuscript. We appreciate for the Reviewer's
kind comments and questions and hope that the reply will meet your approval. Finally,
we would like to express our great appreciation for your valuable suggestions.

**Reviewer 3:**

**Q1:** Self-assembly in Material Science: Indeed, in situ self-assembly has
emerged as a promising way to fabricate porous materials with fine-grained pore
structure at both micron- and nano-scales, that can be achieved by the spontaneous
assembly of small particles driven by chemical reactions. But this is possible in very
specific cases such as shown in:

- ref 23 in the context of lithium battery durability where a facile fabrication of in
situ self-assembled organic/inorganic hybrid compounds with ordered dual-layer
structure was proposed to suppress/disordered/fractal dendritic growth: ordered
organic moieties could in situ self-assemble on the metallic lithium surface hence
forming an ordered dual-layered interphase structure. This is very far from cement
chemistry that proceeds through a dissolution of clinker grains in high pH conditions
followed by the precipitation of the hydrate phase called CSH after a solubility
threshold has been reached.

- In ref 24 of the context of porous polymers with well-defined multiscale pore
morphologies using a solvent-induced self-assembly strategy to synthesize
hierarchical porous polymers with tunable morphology, mesopore and nanopore
structures using the Friedel–Crafts reaction between polymeric compounds and
varying solvent polarity to tailor nanoparticles aggregation. Again, this is completely
different than cement chemistry.

- In ref 25 in the context of porous materials being obtained from templating
copolymeric phases with the idea to eventually obtaining bicontinuous-structured
porous materials. using block copolymers self-assembly with pore sizes and lattice
parameters ranging from 1 to 500 nm. Indeed, this has opened the route to the
synthesis of new families of ordered, pore-size-controlled-by-design porous materials
such as MCMs, SBAs, etc. Again, this is completely different than cement chemistry
as this route will be inoperative in the context of cement paste setting where pH is
only controlled by cement anhydrous clinker grains initial chemistry.

Therefore, although the general idea of guiding cement paste toward a more
controlled texture with pre-ascribed property is certainly appealing, the inherent

complexity of CSH formation casts of lots of doubt on the idea of in situ controlling
the self-assembly of CSH grains in a two-step process that would consist in (i)
constructing a nanoporous hydrogel skeleton (in 10 min??) using rapid gelation and
(ii), cement hydration and setting from the surface of the hydrogel skeleton, gradually
filled-up by hydrogel porosity (filler effect). To me this paper rather reports a simple
quite obvious filling up of the micron size pores of an organic hydrogel (with surface
being hydrophilic with a large OH density per nm²) with CSH grains adsorbing onto
its surface due to obvious oxide particle (CSH) / OH (hydrogel) electrostatic
interactions. Actually, CSH formation was reported as mostly occurring from the
clinker grains surface in general cement chemistry literature through a simple
oxide-on-oxide favorable interaction. Here, this is probably the same situation where
hydrogel surface chemistry is compatible with the growth of CSH.

**(1) It seems to me that the claimed self-assembly is actually largely**
**overstated.** The paper basically shows CSH grains formation starting from the
surface of a hydrogel phase through an obvious adsorption process that is simply due
to the surface chemistry of the initial host matrix. **(2) Nothing really new** as this was
already described in CSH setting with CSH grains adsorbing onto clinker surface as it
gets through dissolution. Can the authors justify the claimed their self-assembly
strategy with cement hydration?

**R1: (1):** Response to ‘**It seems to me that the claimed self-assembly is**
**actually largely overstated**’.

We totally agree with the reviewer's suggestion, after a thorough review of the
referenced literature. The preferable deposition of C-S-H grains on the surface of
hydrogel was driven by the heterogeneous nucleation process^{40,41}, rather than driven
by self-assembly process. Therefore, we deleted the over stated term of
‘self-assembly’ in the revised manuscript.

The intention of using ‘self-assembly’ in the current study was to emphasis the
spontaneously process of the deposition of C-S-H gels on hydrogel skeleton, which
endowed the well-organized porous structure in LLST. Notably, this method of
spontaneously forming porous structures is more convenience than ice template

strategy (Yang Zhou, NC, 2023; Reza Moini, AFM, 2024)^{10, 42}, which requires special
apparatus (directional freezing equipment) and rigorous conditions (including freeze,
thaw and cure process in different temperature). Therefore, this method of
spontaneously forming porous structure shows huge potential for application due to
its simple fabrication process.

**(2): Response to ‘Nothing really new’.**

We acknowledge that the formation of C-S-H gels on the surface of cement
clinker or hydrogel was not new, according to the heterogeneous theory. However, in
the current research, the preferable deposition of C-S-H grains on the surface of
hydrogel not only related to the heterogeneous nucleation sites, but also attributed to
hydrogel providing water for cement hydration, which is reported in the first time.

Specifically, the hydrogel in the current study quickly formed within 10 min after
fabrication, absorbing all the water from the cement slurry and gradually releasing it
for cement hydration. This process facilitated the dissolution of cement particles on
the surface of hydrogel, thus contributing to the preferable deposition of C-S-H grains.
In contrast in previous hydrogel-toughened-cement research, the hydrogel formed
slowly (within 12 ~ 48 h)^{43, 44} and failed to control the water in cement matrix.
Therefore, this unique water controlled mode via hydrogel highlights the novelty of
our study. According to your suggestion, the key roles of gelation rate in inducing of
C-S-H gels precursors on hydrogel skeleton were further discussed in the revised
manuscript.

**Q2: (1) First, the reasons for using hydrogel is not clear at all.** It really
requires digging into the paper to realize that this is in fact an organic phase described
as Poly(NIPAm-SA). **(2) Second, its gelation process is also unclear.** The reader gets
to know that **it involves a solution-to-gel transformation in presence of cement**
**clinker grains.** But the straight definition of the sol-gel transition is the transition
from the sol state (suspension in a liquid) to the gel state (a form of solid). It concerns
materials made up of macromolecular entities capable of bonding from one to the next.
In the sol state, the entities can move individually, whereas in the gel state their bonds

form a three-dimensional network throughout the material. This transition can be
triggered by temperature a decrease or a pH change. It is generally reversible.

It is therefore legitimate to ask whether **this observed gelation only results**
**from the setting of CSH grain onto the organic phase surface** pretty much as in the
carbon-cement compounds (see Pellenq et al, US patent,
<https://patents.google.com/patent/US20190218144A1/en>) where carbon grains are
also together by CSH set phase. **As such, in LLSTs, there will be not real sol-gel**
**transition.** The overall process being controlled by cement setting process that is
accompanied by a large pH increase (12.7) that produces at the same time a CSH
(cement hydrate) phase percolating throughout the initial hydrogel skeleton and
rigidifying it. This is obviously irreversible hence not obeying to the standard sol-gel
transition definition. **(3) Can the authors elaborate mote on LLSTs fabrication.**

**R2: (1):** Response to ‘**the reasons for using hydrogel is not clear at all**’.

Thanks for your kind comments. The intention of using hydrogels is to fully
utilize the uniform micro pores in hydrogels as a template to construct cement paste
with hierarchical porous structure.

Specifically, the formation of hydrogel skeleton quickly constructed a continuous
network throughout the cement matrix, which was composed by micro porous. With
the progress of cement hydration, those micro pores were partially filled by
newly-formed cement hydrates to form nano pores in cement matrix. As a result, the
LLST exhibited a hierarchical structure made up of micro (1~50 μm) and nano pores
(5~100 nm). According to your suggestion, the reasons for using hydrogel were added
in the revised manuscript as follows:

**Line 336: The intention of using hydrogels is to fully utilize the uniform micro**
**pores in hydrogels as a template to construct cement paste with hierarchical porous**
**structure.**

**(2):** Response to ‘**this observed gelation only results from the setting of CSH**
**grain onto the organic phase surface, and in LLSTs, there will be not real sol-gel**
**transition**’.

In fact, the observed solution-to-gel transformation of LLST only referred to the

gelation process of hydrogel, rather than the setting process of C-S-H grain onto the
organic phase surface.

This was because, the LLST exhibited specific elastic behavior within 10 min, as
shown in Fig. 6, during which the cement had not yet begun to hydrate, nor had
C-S-H gels formed (normally after 2 ~ 4 h of hydration ⁴⁵). Therefore, the
solution-to-gel transformation of LLST was not attributed to the cement hydration.
Besides, according to your suggestion, the “sol-gel transition” was inappropriate for
LLST. We changed it to: ‘solution-to-solid transformation’ in the revised manuscript
in Line 375.

**(3): Response to ‘Can the authors elaborate more on LLSTs fabrication’.**

According to your suggestion, the schematic diagram of the gelation process was
added in supplementary file as follows:

[REDACTED]

Figure 6. The fabrication process of LLST.

In addition, the fabrication process of LLST was re-wrote in the revised
manuscript, as follows:

Line 369: Continuous stirring was maintained during the mixing process to
ensure the uniform dispersion of cement particles without sedimentation. During
which the initiator was released by monomer, and then combined with each other to
form the polymer chains. Subsequently, polymer chains were interconnected by

cross-linking agent to form a polymer network. The above gelation process of
hydrogel was completed within 10 minutes as shown in Fig. 7. LLST has undergone a
solution-to-solid transformation and exhibited hydrogel-specific elastic behavior.

**Q3:** Water content: Specifically, a large amount of water seems to be absorbed
by hydrophilic groups in hydrogel skeleton via hydrogen force, which promoted the
in-situ dissolution of cement clinker particles and the precipitation of CSH grains on
the hydrogel skeleton. **(1)** It is quite surprising to see in Fig. 4a that after 28 days,
**there is still clinker C₂S and C₃S phases present. And no trace of CSH in**
**diffractograms.** This is very unexpected as the water to cement ratio as given in Fig.1
show an excess of water: it is usually around 50% for neat OPC paste while here it
varies from 200 to 50% indicating an excess of water in most samples beside that in
sample LLST-4 (w/c=50%). **(2)** This seems to indicate that there is a problem with the
water content management? As water is needed to dissolve anhydrous clinker grains
and achieved CSH precipitation (that occurs at around 20 mmol Ca, concentration),
**what is the rationale behind the water/polymer, water/binder ratios** as cement
grains dissolution and CSH precipitation need the right amount of water given the
hydrogel surface affinity to water?

**R3: (1):** Response to ‘**there is still clinker C₂S and C₃S phases present. And**
**no trace of CSH in diffractograms**’.

Thanks for your comments. The existence of C₂S and C₃S in cement matrix after
28 d was related to the long-term reaction characteristic of cement hydration, as
reported by Barbara Lothenbach et al. ⁴⁶, who found that C₂S and C₃S still existed in
XRD after 1 year ⁴⁷. However, in the current study, the left content of un-hydrated C₂S
and C₃S was only 3% ~ 8%, respectively, based on the QXRD results. These results
indicated the quick reaction rate of clinker in LLST, due to the high w/c ratio. In
addition, the reason behind ‘no trace of CSH in diffractograms’ was attributed to the
amorphous configurations of C-S-H gels, which were hard to be diffracted in XRD.

**(2):** Response to ‘**what is the rationale behind the water/polymer,**

**water/binder ratios’.**

The rationale behind ‘high water/polymer, water/binder ratio’ was not only for
cement grains dissolution and CSH precipitation but also for the introduction of more
pores in cement matrix after water evaporation. In fact, extra water will lead to the
bleeding of cement paste without hydrogel, which makes it difficult to introduce more
pores in cement matrix. However, a large amount of water can be uniformly
distributed or stored in the 3D network of hydrogel, which allows more water
introduced into cement matrix without bleeding. In that case, more pores can be left in
cement matrix after water evaporation. Based on your comments, further explanation
about the high w/c ratio was added in the revised manuscript, as seen in the Line 381.

**Q4:** CSH nucleation theory versus adsorbed water: It is claimed by the authors
that, based on the heterogeneous nucleation theory, the nucleation barrier during the
formation process of cement hydrate CSH grains can be reduced by the nucleation
sites on hydrogel skeleton surface. But at the same time, it is also claimed that
adsorbed water on the same surface facilitates CSH formation. One of the key
statements of this paper is that both of the water and nucleation sites provided by
hydrogel skeleton facilitated the in-situ cement hydration on the hydrogel skeleton.

It is known that CSH grains actually form on clinker grains from the local
variation of pH and ion concentrations due to differences in clinker surfaces energy.
Indeed, clinker grains C₂S, C₃S, are crystals, hence have different surface energies
with respect to the decalcification process that puts Ca ions in solution, depending on
surface crystallographic orientations. Clinker grain surface defects (steps, lacunes...)
can also be the location of CSH precipitation.

But nucleation of CSH grains onto the hydrogel surface relies an obvious catalyst
argument: the surface increasing the encounter probability of reactants. As CSH
formation is a ions-in-solutions process, it seems that adsorbed surface water provides
a way to transport, concentrate and precipitate these species onto the hydrogel surface.

**Why is there a grain precipitation process rather than the formation of a**
**continuous CSH layer all over the hydrogel pore network?**

**R4:** Response to ‘Why is there a grain precipitation process rather than the
formation of a continuous CSH layer’.

Thanks for your kind comments. The grain precipitation of C-S-H gels on
hydrogel network was related to the different roughness of pore walls and intersection
areas in hydrogel.

Normally, the nucleation of C-S-H was considered a two-step process, in which
the globular C-S-H precursor formed in cement solution first, and then transformed
into a foil-like C-S-H. In our research, based on the heterogeneous nucleation theory,
the skeleton of hydrogel played the role of nucleation sites for the formation of C-S-H
precursor. However, the hydrogel skeleton included both smooth, flat areas (pore
walls) and wrinkled regions (intersection areas). According to ^{48,49}, the wrinkled areas
were more conducive to product adhesion. Therefore, the C-S-H grain precipitation in
hydrogel skeleton rather than the formation of a continuous CSH layer, as can be seen
in Fig. R1.

**Fig. R1.** Formation of C-S-H grain on the intersection of hydrogel skeleton

**Q5: (1)** Mechanical properties: The interesting point of the paper is the improved
mechanical properties of LLSTs. **But comparison should be made with neat OPC**
**and not with foam cement pastes** (Fig 3). The 54% reduction of density (compared
to OPC) obviously impacts LLSTs mechanical properties. It has to be weaker than a
straight OPC. Furthermore, density reduction as observed on LLSTs means more
porosity. LLST displayed a 54% reduction in density: this means that LLST has a
density close to that of liquid water. This should be clearly stated and the kind of

porosity range that has changed compared to normal cement paste should be
indicated.

**(2)** The real interesting question though is LLSTs resistance to shear, plastic
deformations and ultimately to fracture (toughness) compared to neat OPC and foam
cement pastes. The authors failed to address these crucial points. **Can the authors**
**provide mechanical information beyond compressive strength such as fracture**
**toughness?**

**R5: (1):** Response to ‘comparison should be made with neat OPC and not
with foam cement’

Thanks for your kind comments. As the reviewer remarked, one of the interesting
points of this manuscript is ‘improved mechanical properties of LLST’, in which
LLST refers to ‘low-carbon, **lightweight**, strong and tough cement-based material’.
Lightweight is an important evaluation index, that is the reason why we compared the
mechanical strength between the LLSTs and other lightweight cement-based materials.
In addition, we agree with that strength and density comparison with neat OPC
(labeled as *Ref.* in this research) is also important, which was emphasized in revised
manuscript:

**Line 147:** Consequently, the proportion of water per unit volume in the cement
product was increased, and led to a reduced density by 11% ~ 43% than *Ref.* (S. 2 (b))
with increased porosity.

**Line 169:** In addition, the specific compressive strength of LLST samples was
37% ~ 145% higher than that of *Ref.*

**(2):** The fracture toughness of LLST was detected based on the recommendation
proposed draft by RILEM Committee on Fracture Mechanics of Concrete-Test
Methods^{50, 51}. The specimens (20×20×80 mm) were cut at mid-span with a half depth,
as shown in **Fig. R2**, and the results were added in supplementary files as S-table 1.

Fig. R2. LLST Sample for fracture toughness test.

S-Table 1. Fracture toughness of LLST.

Specimen	a/W	f(a/W)	Pmax (N)	KIC (MPa·m ^{1/2})	Average (MPa·m ^{1/2})	Improvement (%)
LLST1	0.491	2.589	212.000	0.631	0.609	2.200
	0.497	2.639	204.000	0.619		
	0.486	2.549	197.000	0.577		
LLST2	0.493	2.606	194.000	0.581	0.561	2.027
	0.499	2.656	185.000	0.565		
	0.497	2.639	177.000	0.537		
LLST3	0.504	2.700	178.000	0.552	0.518	1.872
	0.496	2.631	165.000	0.499		
	0.489	2.573	170.000	0.503		
LLST4	0.495	2.623	110.000	0.331	0.398	1.439
	0.498	2.648	144.000	0.438		
	0.506	2.717	136.000	0.425		
Ref.	0.496	2.631	110.000	0.333	0.277	-
	0.498	2.648	87.000	0.265		
	0.492	2.598	78.000	0.233		

Q6: Life cycling Assessment: **(1) This paper lacks of a proper Life cycling Assessment (LCA)** of all involved chemical including the hydrogel phase, the gelation agent, etc... The claim of 64% reduction in CO₂ emission is totally unjustified and probably wrong facing a full LCA assessment giving the used chemicals involving amine that have known and documented health issues.

(2) The disclosed list involves N-isopropylacrylamide (NIPAm), sodium acrylate (SA), N,N-methylenebisacrylamide (Bis), potassium persulfate (APS), and N,N,N',N'-tetramethylethylenediamine (TEMED). **Amines are NOT environmentally friendly.** Can the authors evaluate the overall environment impact of LLSTs?

R6: (1): Response to 'This paper lacks of a proper Life cycling Assessment'.

Based on your suggestion, the life cycle assessment (LCA) of LLST was investigated. The Global Warming Potential (GWP), Acidification Potential (AP) and

994 Eutrophication Potential (EP) of LLST were calculated based on the CO₂, SO₂ and
995 NO_x emission during the raw material production, transportation, and product
processing. The results indicated that GWP, AP and EP of LLST1 were reduced by
49%, 49% and 63%, respectively, as compared with that of Ref. paste (w/c=0.5). The
LCA results confirmed that, compared with conventional cement-based materials,
LLST was environmentally beneficial.

Additionally, CO₂ emission was the most concerned environmental problem. It
should be noted that the CO₂ released by “organic materials production” mainly
comes from the energy use, which could be reduced by using cleaning energy. In
contrast, around 60% of CO₂ released by “cement production” comes from the
decomposition of CaCO₃, which cannot be reduced by using cleaning energy. In that
case, the addition of organic materials to reduce the usage of cement was more
conducive to reducing CO₂ emission.

**(2): Response to ‘Amines are NOT environmentally friendly’.**

Thanks for your kind comments. We admitted that using organic materials,
taking amines as an example, is destructible to environment. However, cement-based
materials have been widely used as solid waste or nuclear sealers. In our research, the
organic materials were encapsulated within the cement matrix, which can effectively
prevent contact between organic materials with environment and reduce the impact on
the environment.

**Q7:** Ab initio metadynamics simulations: The computational part is really naïve
as it does not go beyond a very molecular description of involves moieties using ab
initio technique.

**(1)** The claim “there is currently no suitable force field to simulate the solution
system containing hydrogel and inorganic aqueous species” is wrong. Authors are
advised to get exposed to reactive potential such as ReaxFF that can handle all
suitable elements. There are realistic atomic models of CSH readily available and
probably model for the hydrogel surface that can be handle in the Lammmps simulation
package that implements suitable reactive potentials and different metadynamics

techniques.

(2) Ab initio Metadynamics simulation were performed. But this requires
defining collective variables to obtain free energy profile that is sampling a small
subset of the phase space with no guaranty of sampling the right path. The authors
have chosen only the most obvious one the simple two-body distance. They could
have explored/build other collective variables involving more than just interatomic
pair distance to describe the reactivity between a single Ca and OH hydrogel groups
in presence of water (the density of water is not even specified). Figures 6h and 6i
need proper explanations. The smallness of the simulation system makes the reader
wondering if this is representative of the experimental situation. The art of
metadynamics requires a proper sampling of both sides of energy barrier that hence
needs to be large enough. Did the authors use well-tempered metadynamics? What
energy bias did they use?

(3) à “strong interactions including van der Waals... : DFT ab initio calculations
are notoriously known for their incapacity in predicting van der Waals interactions. In
most ab initio simulation packages, dispersion interactions are included under the
form of an empirical correction. Hence, any force field calculation would be more
efficient as less computationally expensive allowing better sampling of a set of
combination of collective variables. Why not going through this route using for
instance ReaxFF force field that has been used in many instances in the realm of
cement chemistry?

**R7: (1):** Thank you for your comments. We understand your suggestion regarding
the use of reactive force fields like ReaxFF because it can balance between
computational efficiency and the ability to simulate bond formation and breakage.
However, its effective application requires careful consideration of the availability of
parameter sets and training data for parameterization. In fact, ReaxFF relies on
parameter sets tailored to specific combinations of elements and their interactions. In
cement system, the commonly used ReaxFF is developed by Hegoi Manzano and
coworkers, which is only suitable for the Si/Ca/O/H systems (Manzano et al. *Langmuir*
2012, 28 (9), 4187-4197). To the best of our knowledge, there is currently no ReaxFF

force field that can directly and reliably model a system containing all the necessary
elements in our system containing C/H/O/N/Ca/Si. That is why although there are
numerous studies on reactive MD simulations on C-S-H system, there is currently no
study on reactive MD simulation on the organic molecules/ C-S-H system. Another
problem for ReaxFF is that although the aim of this kind of force field is to provide
transferability and be applicable to a broad range of chemical environments, the
parameter sets created for different chemical reactions still exhibit variance due to the
different fitting data. For instance, there are different parameter sets developed for
Mo/S system. One aims to reproduce deformations of MoS₂ sheet (Ostadossein et al.
*The Journal of Physical Chemistry Letters*, 8 (2017) 631-640), and one used to
investigate the reaction between MoO₃ and sulfur to form MoS₂ (Hong et al. *Nano*
*Letters*, 17 (2017) 4866-4872), while the other is to explore crystallization of a single
layer of the Mo-S system at various stoichiometries and oxygen concentrations (Chen
et al. *The Journal of Physical Chemistry C*, 124 (2020) 27571-27579). Nonetheless, if
one wants to simulate the MoS₂ crystallization outside of the single layer, a new
ReaxFF must be developed for the Mo/S system (Ponomarev et al. *The Journal of*
*Physical Chemistry C*, 126 (2022) 9475-9481). Hence, even though the extant
H/O/Ca/Si force field parameters has been successfully applied to numerous systems
of the interface between water and calcium silicate or C-S-H, they may fall short in
faithfully reproducing the other reactions, such as the nucleation of C-S-H from Ca
ions and Si tetrahedra in solution and the dissolution of Ca ions and silicate tetrahedra
from the solid surface. Therefore, we opted for an *ab initio* approach to accurately
simulate the organic/inorganic solution system to capture the detailed interactions
between the Ca ion and the carboxyl group from the hydrogel.

Based on the reviewer's comment, we think our statement regarding to the choice
of the simulation method may be inappropriate. Thus, we remove the content
"Although the classical molecular dynamics (CMD) simulation is appropriate to study
the hydrogel/cement pore solution system with thousands of atoms, and some force
fields, such as the combination of ClayFF⁵² and CVFF⁵³, have been successfully
applied to model the interface between polymer molecules and solid phase of

cement^{54, 55, 56, 57}, there is currently no suitable force field to simulate the solution
system containing hydrogel and inorganic aqueous species.”

To solve the above limitations, we have developed a machine learning force field
(MLFF) for simulation the reaction in this work. MLMD combines the sampling
capabilities of traditional MD with the quantum mechanical accuracy of DFT
calculations. The computational details of the construction of the MLFF were shown in
the manuscript as follows:

[revised manuscript text omitted]

**S. 4. (a) and (b)** Comparison of the energies and atomic forces calculated by the MLFF and DFT
for configurations in the AIMD-based WT-MetaD training dataset. **(c) and (d)** Comparison of the
energies and atomic forces calculated by the MLFF and DFT for configurations in the perturbed
structure training dataset. **(e) and (f)** Comparison of the energies and atomic forces calculated by
the MLFF and DFT for configurations in the test dataset.

**R7: (2):** Thank you for your thoughtful comments and feedback. On the choice
of collective variables (CVs), we agree with the reviewer's suggestion that different
kind of CVs should be tested. Thus, we have explored different kind of CVs. We
should point that it is computationally costly for the ab initio metadynamics
simulations to explore the influence of the CVs on the reaction pathway. Thus, to
solve this problem, we trained a machine learning force field (MLFF) for multiply
metadynamics simulations. based on the MLFF, we choose three set of CVs, which is
the most relevant to the reaction between the Ca ion and carboxyl groups.

For the first CV, we aimed to determine the most stable configuration for the
binding of the Ca ion to the carboxyl group. To achieve this, the coordination numbers
(CN) of the Ca ion with the O ion from water molecules (O_w) along with the CN of
the Ca ion with the O ion from carboxyl group (O_c) were adopted to form the
two-dimensional CVs. For the second CV, as we find that the most stable
configuration of the binding of the Ca ion to the carboxyl group has only one Ca- O_c
bond with five Ca- O_w bond, the distance between the Ca ion and one of the O_c ions
was chosen to explore the reaction mechanism as the Ca ion approaches the carboxyl
group. For the third CV, the distance between the Ca ion and the center of mass (COM)
of two O_c ions, which was initially used in our work, was used to verify the rationality
of the bonding mechanism found by these three different choices of CVs. The
computational setup for the MLMD-based WT-MetaD were provided in the
manuscript as follows:

**Machine learning force field molecular dynamics simulations:** All machine
learning force field molecular dynamics (MLMD) simulations were performed in the
LAMMPS code³⁴ using the fine-tuned model. The periodic boundary conditions were
applied in all directions. The timestep of 1.0 fs was adopted. The initial configuration
was first performed energy minimization with conjugate gradient algorithm and then

equilibrated at the chosen temperature with NPT ensemble for 100 ps. In production
run, the NVT ensemble was applied, and the temperature was controlled by the
Nose-Hoover thermostat with a relaxation time of 100 fs. The duration of the
production run was 1 ns for each simulation.

To comprehensively elucidate the molecular mechanisms underlying the
interaction between the Ca ion and the carboxyl group, three collective variables were
chosen for the WT-MetaD simulations. In the simulation 1 (MLFF 1), we aimed to
determine the most stable configuration for the binding of the Ca ion to the carboxyl
group. To achieve this, the coordination numbers (CN) of the Ca ion with the O ion
from water molecules (O_w) along with the CN of the Ca ion with the O ion from
carboxyl group (O_c) were adopted to form the two-dimensional CVs. The CN was
computed using the following equation (Eq. 2):

$$1199 \quad CN(Ca - O_{w/c}) = \sum_{j \in O_{w/c}} s_{ij}(r_{ij}) = \sum_{j \in O_{w/c}} \frac{1 - \left(\frac{r_{ij} - d_0}{r_0}\right)^n}{1 - \left(\frac{r_{ij} - d_0}{r_0}\right)^m} \quad (2)$$

where r_{ij} is the distance between atom i and atom j . $s_{ij}(r_{ij})$ is a switching function
describing the coordination between atom i and j . d_0 is the central value of the
function. r_0 is the acceptance distance of the switching function. The d_0 and r_0 were set
as 2.42 Å and 0.4 Å, respectively, which were tested to be appropriate^{35, 36}. n and m
are 6 and 12, respectively^{37, 38}.

In simulation 2 (MLFF 2), the distance between the Ca ion and one of the O_c ions
was chosen to explore the reaction mechanism as the Ca ion approaches the carboxyl
group. In simulation 3 (MLFF 3), the distance between the Ca ion and the center of
mass (COM) of two O_c ions was selected to verify the rationality of the bonding
mechanism found by different choice of CVs. The parameters of the MLFF-based
WT-MetaD are shown in Table 2. Convergence tests for free energy surfaces, time
evolution of the CVs, time evolution of the height of bias are shown in Fig. S. 5-7.
The errors in the free energies for MLFF 2 and MLFF 3 were calculated using the
block-analysis method³⁹. The average error converged at the block size of around 900
in both simulations (Fig. S. 8). Using 900 blocks, the average errors were calculated
to be 1.22 kJ/mol for MLFF 2 and 1.42 kJ/mol for MLFF 3.

Table 2. Parameters of WT-MetaD simulations

Project	CV	Quadratic walls		Initial height	Width	Deposition frequency	Biasfactor
		Lower position	Upper position				
AIMD	Distance	1.8 Å	6.8 Å		0.1 Å		
MFL1	CN(Ca-O _c)	3.5		3.5	0.1	30 fs	20
	CN(Ca-O _w)	0.1			0.05		
MFL2	Ca-O _c Distance	2 Å	6.5 Å	kJ/mol	0.1 Å		
MFL2	Ca-COM Distance	2 Å	5.5 Å		0.07 Å		

1217

1218 The results of the MLMD-based metadynamics were also shown in the
1219 manuscript as follows:

[revised manuscript text omitted]

For the density of water, we are sorry not explicitly mention this information.
 The water density is approximately $1g/cm^3$ in this work. We also supply this
 information in the manuscript as follows:

The NIPAM-SA monomer, together with a single Ca ion, is placed within an
 explicit solvation shell with a hydration radius of 5.5 \AA (the density of water is
 $1g/cm^3$).

For the improper explanations for the IRI analysis, we rewrite this part in the
 manuscript as follows:

To characterize the nature of the Ca-hydrogel interaction, we performed an
 interaction region analysis (IRI) analysis (Figs. 5(e)), a novel real-space function that

identifies both weak interaction and chemical bond. The IRI analysis reveals a blue
region between the Ca ion and one of the O_c ions, indicating a strong electrostatic
attraction and the formation of a chemical bond. An orange region occurs between the
Ca ion and center of carboxyl group suggests a steric hindrance effect. These findings
are consistent with our experimental observations.

On the size of the simulation system, as we aim to specifically simulate the
molecular mechanisms of the interaction between the Ca ion and the carboxyl group of
the hydrogel, the number of the concerned atoms in this reaction is not large. When we
construct the model, a 5.5 Å hydration shell radius of the Ca-hydrogel system was used.
This aims to provide enough water molecules that can prevent the self-interaction due
to the image effect caused by the periodic boundary condition in the MD simulations.
The 5.5 Å radius of the hydration shell for Ca-hydrogel is large enough for the model to
consider the solvent effects. A larger shell would significantly increase computational
costs without contributing much additional accuracy for this system. Besides, the
current model, comprising 481 atoms within an orthogonal simulation box of
dimensions 16.6 Å × 17.2 Å × 17.6 Å, is nearly the up limit for the *ab initio* molecular
dynamics simulations. Thus, we believe this system with the most advanced
simulation method can ensure that the molecular mechanisms we uncover are reliable
and reflective of the underlying physics and chemistry.

On the details of the metadynamics, we are sorry that we did not mention the
version of the metadynamics we used in this work. In this work, we indeed used the
well-tempered metadynamics (WT-MetaD) method. The detailed parameters are
shown in the Table 2 of the manuscript (shown in the above). Thanks to the MLFF, both
sides of energy barrier has been sampled properly under a 1 ns long simulation time.
The sampling situation can be checked by the time evolution of the CV (Figs. S5abd,
S6ac, S7ac, shown below). The information of the energy bias we used are also shown
in the Table 2. The height of the energy bias is initially set as 3.5 kJ/mol, which will
decrease with time (Figs. S5c, S6b, S7b). The width of the energy bias is set to be 0.1,
0.05 in the MLFF 1, 0.01 Å in the MLFF 2, and 0.07 in the MLFF 3, respectively.

**S. 5** (a) and (b) Convergence tests for free energy surfaces as a function of $\text{CN}(\text{Ca-O}_w)$ and
 $\text{CN}(\text{Ca-O}_c)$, which is performed every 30 ps (1000 Gaussian kernels deposited) along the 1 ns
 simulation time. The metadynamics is converged after 1 ns. (c) Time evolution of the height of the
 bias added to the system. The height of the bias decreases with the time. (d) Time evolution of
 $\text{CN}(\text{Ca-O}_w)$ and $\text{CN}(\text{Ca-O}_c)$ during the simulation time.

**S. 6** (a) Convergence tests for free energy surfaces as a function of Ca-O_c distance, which is
 performed every 30 ps (1000 Gaussian kernels deposited) along the 1 ns simulation time. The
 metadynamics is converged after 1 ns. (b) Time evolution of the height of the bias added to the
 system. The height of the bias decreases with the time. (c) Time evolution of Ca-O_c distance
 during the simulation time.

**S. 7 (a)** Convergence tests for free energy surfaces as a function of Ca-COM distance, which
 is performed every 30 ps (1000 Gaussian kernels deposited) along the 1 ns simulation time. The
 metadynamics is converged after 1 ns. **(b)** Time evolution of the height of the bias added to the
 system. The height of the bias decreases with the time. **(c)** Time evolution of Ca-COM distance
 during the simulation time.

**R7: (3):** For the description of the van der Waals (vdW) interaction, we agree
 with the reviewer that the DFT method with the traditional exchange-correlation
 functionals, such as B3LYP and PBE, are poor in calculating dispersion effect. To
 solve this problem, we have adopted the most popular dispersion correction method,
 the Grimme's D3 method (DFT-D3), to correct the total energy, energy gradient and
 frequencies due to the London-dispersion interactions in the system (Grimme et al,
 *The Journal of chemical physics* 132, 154104 (2010)). This method introducing a
 vdW-dispersion energy-correction term can credibly increase the accuracy of
 describing the vdW-dispersion effect (Lu et al. *J Mol Model* 19, 5387–5395 (2013)).
 In fact, since the advent of DFT-D (Density Functional Theory with dispersion
 corrections), it has transformed into one of the most effective methods for calculating
 weak interactions. In contrast, ReaxFF uses a bond length–bond order scheme to
 describe atomic interactions, it is hard to admit this kind of method can give a better
 description for the vdW interaction than DFT-D method.

The reason why we use the *ab initio* method to simulate the interaction between
 Ca and the carboxyl group from the hydrogel is that this method can involve the
 electrons of the atoms to give an accurate description of the atomic behavior as the
 essence of the chemical reaction is the transfer of the electrons between atoms. Neither
 the traditional MD nor the reactive MD can directly involve the electrons in the system.
 Besides, there is no suitable ReaxFF potential for the C/H/O/N/Ca/Si system. The only

one ReaxFF for cement can only describe the system contain H/O/Ca/Si elements. And
the parameters of this ReaxFF were fitted using DFT calculations on gas phase
calcium–water clusters, calcium oxide bulk and surface properties, calcium hydroxide,
bcc and fcc Ca, and proton transfer reactions in the presence of calcium. It is definably
less accurate than *ab initio* simulation regarding to the interaction between atoms. The
only demerit of the *ab initio* method is that it is computationally expensive, so that it is
impossible to test different collective variables for one reaction and maybe an
insufficient sampling especially when using enhanced sampling method.

To solve the above limitations, we have developed a MLFF for simulation the
reaction in this work. MLMD combines the sampling capabilities of traditional MD
with the quantum mechanical accuracy of DFT calculations.

In summary, we tried our best to improve the manuscript and made some
changes with remarks in the revised manuscript. We appreciate for the Reviewer's
kind comments and questions and hope that the reply will meet your approval. Finally,
we would like to express our great appreciation for your valuable suggestions.

**Reference:**

- 1. Andreas Picker LN, Helmut Cölfen. Mesocrystalline calcium silicate hydrate: A bioinspired route
toward elastic concrete materials. *Science advances* **3**, 1701216 (2017).
- 2. Marchon D, Flatt RJ. Impact of chemical admixtures on cement hydration. In: *Science and*
*Technology of Concrete Admixtures* (2016).
- 3. Flatt RJ, Roussel N, Bessaies-Bey H, Caneda-Martínez L, Palacios M, Zunino F. From physics to
chemistry of fresh blended cements. *Cement and Concrete Research* **172**, (2023).
- 4. Marchon D, Juilland P, Gallucci E, Frunz L, Flatt RJ. Molecular and submolecular scale effects of
comb-copolymers on tri-calcium silicate reactivity: Toward molecular design. *Journal of the American*
*Ceramic Society* **100**, 817-841 (2017).
- 5. Dalas F, Pourchet S, Rinaldi D, Nonat A, Sabio S, Mosquet M. Modification of the rate of
formation and surface area of ettringite by polycarboxylate ether superplasticizers during early
C3A–CaSO₄ hydration. *Cement and Concrete Research* **69**, 105-113 (2015).
- 6. Palacios M, *et al.* Heating cement to slow down its hydration: The unexpected role of PCE
interpolymer bridge formation. *Cement and Concrete Research* **156**, (2022).
- 7. Wang H, *et al.* Deciphering the influence of superabsorbent polymers on cement hydration and
portlandite formation. *Construction and Building Materials* **418**, (2024).
- 8. Wang H, Lu Z, Wang F, Li Y, Ou Z, Jiang J. A novel strategy to reinforce double network
hydrogels with enhanced mechanical strength and swelling ratio by nano cement hydrates. *Polymer*
**269**, (2023).
- 9. Sun G, Li Z, Liang R, Weng LT, Zhang L. Super stretchable hydrogel achieved by non-aggregated
spherulites with diameters <5 nm. *Nat Commun* **7**, 12095 (2016).
- 10. Chen Y, *et al.* Multi-layered cement-hydrogel composite with high toughness, low thermal
conductivity, and self-healing capability. *Nat Commun* **14**, 3438 (2023).
- 11. Chen G, *et al.* Bioinspired 3D Printing of Functional Materials by Harnessing Enzyme-Induced
Biomineralization. *Advanced Functional Materials* **32**, (2022).
- 12. Haase NR, Shian S, Sandhage KH, Kröger N. Biocatalytic Nanoscale Coatings Through
Biomimetic Layer-by-Layer Mineralization. *Advanced Functional Materials* **21**, 4243-4251 (2011).
- 13. Wang W, Chen SJ, Chen W, Duan W, Lai JZ, Sagoe-Crentsil K. Damage-tolerant material design
motif derived from asymmetrical rotation. *Nat Commun* **13**, 1289 (2022).
- 14. Han D, Ferron RD. Influence of high mixing intensity on rheology, hydration, and microstructure
of fresh state cement paste. *Cement and Concrete Research* **84**, 95-106 (2016).
- 15. Wang H, *et al.* Assessment of the performances and reactions of quaternary LC2–slag cement.
*Advances in Cement Research* **34**, 529-541 (2022).
- 16. Qi C, Jiang F, Yang S. Advanced honeycomb designs for improving mechanical properties: A
review. *Composites Part B: Engineering* **227**, (2021).
- 17. Al-Hasani LE, *et al.* Quantifying concrete adiabatic temperature rise based on
temperature-dependent isothermal calorimetry; modeling and validation. *Materials and Structures* **55**,
(2022).
- 18. Hernandez-Bautista E, Bentz DP, Sandoval-Torres S, de Cano-Barrita PF. Numerical simulation of
heat and mass transport during hydration of Portland cement mortar in semi-adiabatic and steam curing
conditions. *Cem Concr Compos* **69**, 38-48 (2016).

- 19. Zhou J, Ye G, van Breugel K. Characterization of pore structure in cement-based materials using
pressurization–depressurization cycling mercury intrusion porosimetry (PDC-MIP). *Cement and*
*Concrete Research* **40**, 1120-1128 (2010).
- 20. VandeVondele J, Krack M, Mohamed F, Parrinello M, Chassaing T, Hutter J. Quickstep: Fast and
accurate density functional calculations using a mixed Gaussian and plane waves approach. *Computer*
*Physics Communications* **167**, 103-128 (2005).
- 21. Goedecker S, Teter M, Hutter J. Separable dual-space Gaussian pseudopotentials. *Physical Review*
*B* **54**, 1703 (1996).
- 22. Hartwigsen C, Goedecker S, Hutter J. Relativistic separable dual-space Gaussian pseudopotentials
from H to Rn. *Physical Review B* **58**, 3641 (1998).
- 23. VandeVondele J, Hutter J. Gaussian basis sets for accurate calculations on molecular systems in
gas and condensed phases. *The Journal of chemical physics* **127**, (2007).
- 24. Perdew JP, Burke K, Ernzerhof M. Generalized gradient approximation made simple. *Physical*
*review letters* **77**, 3865 (1996).
- 25. Grimme S, Antony J, Ehrlich S, Krieg H. A consistent and accurate ab initio parametrization of
density functional dispersion correction (DFT-D) for the 94 elements H-Pu. *The Journal of chemical*
*physics* **132**, 154104 (2010).
- 26. Claverie J, Bernard F, Cordeiro JMM, Kamali-Bernard S. Ab initio molecular dynamics
description of proton transfer at water-tricalcium silicate interface. *Cement and Concrete Research* **136**,
106162 (2020).
- 27. Leung K, Remppe SB. Ab initio rigid water: Effect on water structure, ion hydration, and
thermodynamics. *Physical Chemistry Chemical Physics* **8**, 2153-2162 (2006).
- 28. Bussi G, Donadio D, Parrinello M. Canonical sampling through velocity rescaling. *The Journal of*
*Chemical Physics* **126**, (2007).
- 29. Bonomi M, Parrinello M. Enhanced sampling in the well-tempered ensemble. *Physical review*
*letters* **104**, 190601 (2010).
- 30. Tribello GA, Bonomi M, Branduardi D, Camilloni C, Bussi G. PLUMED 2: New feathers for an
old bird. *Computer Physics Communications* **185**, 604-613 (2014).
- 31. Zeng J, et al. DeePMD-kit v2: A software package for deep potential models. *The Journal of*
*Chemical Physics* **159**, (2023).
- 32. Wang H, Zhang L, Han J, Weinan E. DeePMD-kit: A deep learning package for many-body
potential energy representation and molecular dynamics. *Computer Physics Communications* **228**,
178-184 (2018).
- 33. Zhang D, et al. DPA-2: Towards a universal large atomic model for molecular and material
simulation. *arXiv preprint arXiv:231215492*, (2023).
- 34. Thompson AP, et al. LAMMPS-a flexible simulation tool for particle-based materials modeling at
the atomic, meso, and continuum scales. *Computer Physics Communications* **271**, 108171 (2022).
- 35. Li Y, Pan H, Liu Q, Ming X, Li Z. Ab initio mechanism revealing for tricalcium silicate
dissolution. *Nature Communications* **13**, 1253 (2022).
- 36. Li Y, Pan H, Li Z. Unravelling the dissolution dynamics of silicate minerals by deep learning
molecular dynamics simulation: A case of dicalcium silicate. *Cement and Concrete Research* **165**,
107092 (2023).

- 37. Li Y, Sun Z, Li Z, Chen B, Li Z. Dimeric and oligomeric interactions between calcium silicate
aqua monomers before calcium silicate hydrate nucleation. *Cement and Concrete Research* **173**,
107297 (2023).
- 38. Li Y, Pan H, Li Z. Ab initio metadynamics simulations on the formation of calcium silicate aqua
complexes prior to the nucleation of calcium silicate hydrate. *Cement and Concrete Research* **156**,
106767 (2022).
- 39. Bussi G, Laio A. Using metadynamics to explore complex free-energy landscapes. *Nature Reviews*
*Physics* **2**, 200-212 (2020).
- 40. Mackay M, *et al.* General strategies for nanoparticle dispersion. *Science* **311**, 1740-1743 (2006).
- 41. Yang J, *et al.* Probing structure–heterogeneous nucleation efficiency relationship of mesoporous
particles in polylactic acid microcellular foaming by supercritical carbon dioxide. *Journal of*
*Supercritical Fluids* **95**, 228-235 (2014).
- 42. Gupta S, Esmaceli HS, Moini R. Tough and Ductile Architected Nacre-Like Cementitious
Composites. *Advanced Functional Materials*, (2024).
- 43. Peiliang C, Changhao L, Zhiyu H, Yuanfeng Z. A comprehensive review on polyurethane
modified asphalt: Mechanism, characterization and prospect. *Journal of Road Engineering* **3**, 315-335
(2023).
- 44. Chen S, *et al.* An experimental and theoretical study of biomimetic cement-epoxy resin
composites: structure, mechanical properties, and reinforcement mechanisms. *Composites Part A:*
*Applied Science and Manufacturing* **185**, (2024).
- 45. Ylmén R, Jäglid U, Steenari B-M, Panas I. Early hydration and setting of Portland cement
monitored by IR, SEM and Vicat techniques. *Cement and Concrete Research* **39**, 433-439 (2009).
- 46. Lothenbach B, Le Saout G, Gallucci E, Scrivener K. Influence of limestone on the hydration of
Portland cements. *Cement and Concrete Research* **38**, 848-860 (2008).
- 47. Jensen OM, Hansen PF. Water-entrained cement-based materials I. Principles and theoretical
background. *Cement and Concrete Research* **31**, 647-654 (2001)
- 48. Lu Z, *et al.* Early-age interaction mechanism between the graphene oxide and cement hydrates.
*Construction and Building Materials* **152**, 232-239 (2017).
- 49. Zhao L, *et al.* Investigation of the effectiveness of PC@GO on the reinforcement for cement
composites. *Construction and Building Materials* **113**, 470-478 (2016).
- 50. Determination of the fracture energy of mortar and concrete by means of three-point bend tests on
notched beams. *Materials and Structures* **18**, 287–290 (1985).
- 51. Li W-w, *et al.* Electrochemical impedance interpretation for the fracture toughness of carbon
nanotube/cement composites. *Construction and Building Materials* **114**, 499-505 (2016).
- 52. Cygan RT, Liang J-J, Kalinichev AG. Molecular models of hydroxide, oxyhydroxide, and clay
phases and the development of a general force field. *The Journal of Physical Chemistry B* **108**,
1255-1266 (2004).
- 53. Dauber-Osguthorpe P, Roberts VA, Osguthorpe DJ, Wolff J, Genest M, Hagler AT. Structure and
energetics of ligand binding to proteins: Escherichia coli dihydrofolate reductase-trimethoprim, a
drug-receptor system. *Proteins: Structure, Function, and Bioinformatics* **4**, 31-47 (1988).
- 54. Chen Y, *et al.* Multi-layered cement-hydrogel composite with high toughness, low thermal
conductivity, and self-healing capability. *Nature Communications* **14**, 3438 (2023).

- 55. Hou D, Xu J, Zhang Y, Sun G. Insights into the molecular structure and reinforcement mechanism
of the hydrogel-cement nanocomposite: an experimental and molecular dynamics study. *Composites*
*Part B: Engineering* **177**, 107421 (2019).
- 56. Hou D, Yu J, Wang P. Molecular dynamics modeling of the structure, dynamics, energetics and
mechanical properties of cement-polymer nanocomposite. *Composites Part B: Engineering* **162**,
433-444 (2019).
- 57. Qiao G, Hou D, Wang P, Lu Z. Insights on failure modes of calcium-silicate-hydrate interface
strengthened by polyacrylamides: Structure, dynamic and mechanical properties. *Construction and*
*Building Materials* **278**, 122406 (2021).
- 58. Jalilehvand F, Spångberg D, Lindqvist-Reis P, Hermansson K, Persson I, Sandström M. Hydration
of the calcium ion. An EXAFS, large-angle X-ray scattering, and molecular dynamics simulation study.
*Journal of the American Chemical Society* **123**, 431-441 (2001).

Reviewer #2:

The authors' revision works satisfactorily addressed all my comments. Based on this consideration, I recommend this manuscript to be accepted for publication.

Response: Thank you for taking the time to review our manuscript. We are delighted that our revisions have satisfactorily addressed your comments. Your valuable feedback has significantly improved the quality of our work, and we are grateful for your thoughtful insights throughout the review process.

Reviewer #3:

The authors have adequately answered my comments. However, I still have the following questions on the accuracy of the metadynamics simulations in regard to the provided data.

Q1: Although it is true that the well-tempered metadynamics ensures convergence of the free energy by reducing the height of the gaussians, this is a necessary condition, but it is not sufficient to guarantee that the calculations are converged. In fact, one need to monitor the number of barrier re-crossings events to show that both sides of the barrier are adequately sampled. A bias-factor of 20 pretty large (see table), hence very long simulation is required to converge the calculations. Can the authors give a convincing argument re the quality/convergence of their simulations?

R1: Thank you for acknowledging our previous responses. We appreciate your continued interest in the accuracy and convergence of our metadynamics simulations. We fully agree that the reduction in Gaussian height in well-tempered metadynamics is a necessary but not sufficient condition for convergence. Therefore, when we checked the convergence of the well-tempered metadynamics, we have employed multiple criteria to rigorously assess the convergence of our simulations.

Firstly, the most critical and direct indicator of convergence is the stability of the free energy surface over time. Specifically, using the simulation with CVs of CN(Ca-O_w) and CN(Ca-O_c) as an example (Fig. S5), we monitored the time evolution of the free energy surface every 1000 Gaussian bias deposited (given the extensive number of curves calculated, the data prior to 480 ps was omitted for clarity). This assessment is grounded in the fundamental relationship between the free energy and the accumulated bias potential, as described by the following equation:

$$V(s, t \rightarrow \infty) = - \left(1 - \frac{1}{\gamma}\right)F(s) + C$$

Once the free energy surface has converged, it remains unchanged, except for a constant shift in its absolute value. Figs. S5(a) and S5(c) illustrate the evolution of the free energy surface, plotted every 1,000 Gaussian bias potentials deposited over the 1 ns simulation. As shown, there is no observable difference between the free energy surfaces at 960 ps and 990 ps, providing strong evidence of convergence in our simulations.

Additionally, it is necessary to observe the height of the Gaussian bias to reduce when the free energy surface approaches convergence. Initially, the Gaussian heights are larger when the system explores new regions of the phase space. However, as the bias fills the local minima on the free energy surface, the Gaussian heights decrease. A persistently small Gaussian height indicates that no new conformational space is being explored. This behavior is further corroborated by examining the CVs to confirm that all meaningful regions of the CV space have been sampled. In this study, Fig. S5(c) demonstrates the reduction in Gaussian height over the course of the simulation, while Fig. S5(d) confirms that all relevant and meaningful CV values have been adequately explored.

Moreover, as recommended by the reviewer, it is essential to ensure that both sides of the free energy barrier are adequately sampled, which can be verified by observing the diffusion of CV values across the barrier. Fig. 5(a) reveals four minima in the CN(Ca-O_w) distribution at values of 3.5, 4, 5, and 6. Fig. 5(d) (blue dots) shows that the CN(Ca-O_w) values diffuse smoothly between these minima multiple times during the simulation. Similarly, Fig. 5(b) identifies three minima for CN(Ca-O_s) at values of 0, 1, and 2, and the gray dots in Fig. 5(d) indicate that CN(Ca-O_s) values also diffuse smoothly across these regions multiple times. In summary, We believe the free energy surface obtained from the well-tempered metadynamics simulations in this work has rigorously converged.

For the bias factor, its primary function is to regulate the rate of reduction in Gaussian height. An appropriately chosen bias factor prevents the system from exploring unreasonable configurational spaces while ensuring adequate sampling in regions with

meaningful CV values. Specifically, the height of the Gaussian bias potential is determined by the following equation:

$$W(t) = W_0 \exp\left(-\frac{V(s, t)}{k_B \Delta T}\right)$$

where $W(t)$ is the height of the Gaussian bias at time t , W_0 is the initial Gaussian height, $V(s, t)$ is the accumulated bias potential at the collective variable position s and time t , k_B is the Boltzmann constant, and ΔT is a parameter with the dimensions of temperature, referred to as the bias factor (γ) in the following equation:

$$\gamma = \frac{T + \Delta T}{\Delta T}$$

A larger bias factor (γ) corresponds to a higher ΔT , which slows a slower reduction in the height of the added Gaussian potentials. In this work, the chosen bias factor is appropriate, as it results in a suitable reduction rate for the Gaussian potential height (the Gaussian height can reduce to very low before end of the simulation). When the Gaussian height decreases to a small value, it indicates that the configurational space has been thoroughly explored. At this condition, the CVs can diffuse smoothly across both sides of the barrier.

Besides, we think there is no universally accepted definition of “long simulation time”, while there are well-established criteria for assessing convergence. For ab initio-based molecular dynamics simulations, a 1 ns trajectory represents a considerable computational effort and definitely can be regarded as a long simulation time. We believe that once the above conditions—thorough configurational exploration, CV diffusion, and a stable free energy surface ,etc.—are satisfied, the metadynamics simulations can be confidently considered converged.

S. 5 (a) and (b) Convergence tests for free energy surfaces as a function of CN(Ca-O_w) and CN(Ca-O_c), which is performed every 30 ps (1000 Gaussian kernels deposited) along the 1 ns simulation time. The metadynamics is converged after 1 ns. (c) Time evolution of the height of the bias added to the system. The height of the bias decreases with the time. (d) Time evolution of CN(Ca-O_w) and CN(Ca-O_c) during the simulation time.

Q2: Moreover, it is very important that the reactant states don't change throughout the simulation to ensure that after each barrier crossing the same states are sampled. It is also important that the recrossing happens along the same pathway so that the corresponding barrier is explored. (see ref 69 in the present manuscript). Given the fact that atoms may significantly rearrange after the Ca has found a O_w, the free energies calculated here probably have very large errors, that cannot be quantified. It would be revealing to repeat the same metadynamics calculation for one Ca site a number of times using different starting condition, to demonstrate the variability of each data point. Can the authors comment this point in regard to their system?

R2: Thank you for your comments. Following the reviewer's suggestion, we revisit the ref 69 (Bussi and Laio, *Nature Reviews Physics* 2, 200-212 (2020)). We concur that the recrossing needs to happen along the same pathway to explore the barrier. However, based on our understanding, the "same pathway" refers to the path connecting the reactant and product states as defined by the CV, rather than requiring the atomic positions of all atoms in the system at reactant and product states to remain fixed throughout the simulation. Upon re-examining the reference, we did not find any explicit statement indicating that the atomic positions of the system should be

constrained to remain unchanged during the simulation. Instead, we believe that the consistency of the pathway is indicated by the CV values, which are specifically designed to capture the essential degrees of freedom relevant to the reaction or transition under study. In this context, the CV serves as the descriptor of the pathway, not the atomic coordinates.

Metadynamics is a kind of enhanced sampling method that modifies the system's energy landscape to facilitate crossing energy barriers. By design, it does not impose constraints on atomic positions but instead relies on the system's natural ability to explore various configurations within the CV-defined space, which guarantees the minimum artifacts introduced to the reaction pathway. This approach inherently accounts for atomic rearrangements, which are crucial for capturing the entropy contributions to the free energy of the system. For instance, the rearrangements of atoms, including those beyond the Ca and coordinated O_w atoms, are essential to accurately compute the free energy, particularly its entropic components.

Chemical reactions are inherently characterized by dimensionality reduction, where the high-dimensional atomic positions are projected onto a lower-dimensional space defined by the CVs. Metadynamics simulations explore the entire accessible configurational space based on parameters such as the bias factor and initial Gaussian height. Importantly, the exploration is independent of the initial configuration, provided that the CV of the starting structure is chemically meaningful and falls within the relevant region of the free energy surface. Consequently, the starting configurations have minimal influence on the resulting free energy surface. Once convergence is achieved, the relative positions and depths of the minima on the free energy surface, as well as their differences, remain consistent. This demonstrates that the free energy surface is robust and independent of the initial configuration, provided that proper sampling criteria are met. Thanks for your kind comments, we will pay more attention to this question in future research.

Q3: The Gaussians' deposition rate is very fast 30 fs, the system would have no chance of relaxing to the new equilibrium state. Although this a common practice for ab-initio MD simulations, it makes the results quite unreliable. A deposition rate in the range 1-2ps would be more appropriate at least for classical MD simulation part. With the

chosen dual collective variable, convergence might be hard to get over 200 ps trajectories.

R3: Thank you for your constructive feedback. We acknowledge that a Gaussian deposition rate of 30 fs is relatively fast when compared to the classical MD simulations. However, this choice was made to accommodate the computational expense of *ab initio* MD simulations. The faster deposition rate is indeed a common practice in AIMD studies, as it balances the computational cost with the need for enhanced sampling within the limited simulation timescales typically achievable in such calculations. Additionally, we would like to clarify that our simulation ran for 1 ns, not 200 ps as mentioned by the reviewer. This extended simulation time, which is notably long for AIMD, provides confidence that our results are both converged and reliable. The combination of the long simulation duration and the precision of the AIMD approach ensures that the key regions of the free energy surface have been adequately explored, leading to robust conclusions.

Besides, we need to point out that every configuration during AIMD simulations is optimized through density functional theory (DFT) calculations. These calculations determine atomic positions, bond lengths, and other structural properties based on the electronic structure of the system, ensuring high precision and accuracy at each step. While Gaussian bias is added every 30 steps, the electronic structure calculations enable the system to adjust quickly and accurately, maintaining the physical reliability of the simulated configurations.

In contrast, classical MD simulations rely on predefined force fields with fitted parameters, which often lack the accuracy required to fully capture the intricate details of atomic interactions and structural information. As a result, classical MD simulations need a longer deposition interval for equilibrium. Besides, even if a longer deposition interval was used in MD simulations, which seems to make the results more reliable than AIMD, the atomic configurations may still be unreasonable due to the limitations of the accuracy of the fitted force field. This may also lead to unreliable results for MD based metadynamics despite the additional relaxation time. For example, the coordination number of the Ca ion in solution is typically inappropriate in classical MD simulations. Therefore, although the deposition rate in AIMD simulations is faster, the high precision of DFT ensures that the generated configurations are both accurate and

physically meaningful. This provides confidence in the reliability of our results. In the future research, we will follow the reviewer's suggestion to decrease the deposition rate to consistently increase the accuracy of the AIMD-based metadynamics simulations.

Q4: I urge the authors to have a look to the work by Silvestri et al. (J Chem Theory Comput. 2022 Oct 11;18(10):5901-5919. doi: 10.1021/acs.jctc.2c00787). Silvestri et al. did not propose a method to increase the efficiency of the calculations, but they suggested a protocol to ensure that the calculated free energy was accurate and, more importantly, referenced to standard conditions, as the experiment should be. Silvestri et al. propose to use a linear cylindrical restraint to ensure that the volume of the bound and unbound states are well-defined, which means it is possible to know what the reference state of the calculation is. The free energies contributions to add and remove the restraints also need to be computed. Can the authors position their work with respect to Silvestri's scheme?

R4: Thank you for bringing the work of Silvestri et al. to our attention. In their study, they compute the free energies for ion removal from kink sites at the solid-liquid interface. Their objective is to correct the thermodynamics of ion transfer between the solid and solution for a given definition of energy and forces, rather than achieving absolute accuracy relative to experimental results. In their metadynamics simulations, Silvestri et al. use several sets of CVs: the distance from the kink site, the height of the ion relative to its original equilibrium kink site position, and the combination of height and coordination number. To restrain the CV space, they employ a two-step approach: an interface-parallel harmonic wall at ~ 15 Å from the surface was firstly used to restrain the distance of the ion normal to the surface, and the position in the plane parallel to the interface was restrained with a harmonic cylindrical wall. This setup is analogous to funnel metadynamics, which was originally developed for the quantitative calculation of protein–ligand binding free energy. The development of funnel metadynamics is motivated by the challenges faced in simulations of ligand binding with the protein, where once the ligand leaves the binding pocket, it has difficulty finding its way back and starts exploring all of the possible solvated states. These conformations represent a vast part of the configuration space that cannot be sampled thoroughly in a limited computation time. Consequently, once out, the ligand does not again find the binding site, and multiple binding/unbinding events, which are the key to an accurate determination of the binding free energy in metadynamics, cannot be

observed. Silvestri et al. adopt a similar approach to ensure that the dissolved ion can return to its undissolved state on the surface efficiently.

In comparison, our study uses a different set of CVs, specifically the coordination numbers of Ca with O_w and O_c , rather than distances. The use of coordination numbers inherently ensures that the distance between the Ca ion and the carboxyl group does not become excessively large. This avoids scenarios where after O_c reaches a value of 0, it cannot return to 1 or 2, thereby ensuring the diffuse of CV in all relevant configurational space (Fig. S5(d)). Therefore, our approach guarantees proper sampling in the concerned regions without the need for additional restraining like those used by Silvestri et al. Furthermore, we have accounted for the influence of walls in our free energy calculations, and the free energy surfaces presented in our manuscript have been appropriately corrected to reflect these effects.

In summary, thanks for your kind comments and questions. We hope our response has adequately addressed your concerns regarding the accuracy of the metadynamics simulations. Thank you again for your valuable and insightful comments.